# Beyond Tokens: Enhancing RTL Quality Estimation via Structural Graph Learning

Yi Liu [1]   Hongji Zhang [1]   Yiwen Wang [2]   Dimitris Tsaras [2]   Lei Chen [2]   Mingxuan Yuan [2]   Qiang Xu [1]

## Abstract

Estimating the quality of register transfer level (RTL) designs is crucial in the electronic design automation (EDA) workflow, as it enables instant feedback on key performance metrics like area and delay without the need for time-consuming logic synthesis. While recent approaches have leveraged large language models (LLMs) to derive embeddings from RTL code and achieved promising results, they overlook the structural semantics essential for accurate quality estimation. In contrast, the control data flow graph (CDFG) view exposes the design's structural characteristics more explicitly, offering richer cues for representation learning. In this work, we introduce StructRTL, a novel structure-aware graph self-supervised learning framework for improved RTL design quality estimation. By learning structure-informed representations from CDFGs, StructRTL significantly outperforms prior art on various quality estimation tasks. To further boost performance, we incorporate a knowledge distillation strategy that transfers low-level insights from post-mapping netlists into the CDFG-based predictor. Experimental results demonstrate that StructRTL establishes new state-of-the-art results, highlighting the effectiveness of combining structural learning with cross-stage supervision.

## 1. Introduction

As AI models continue to grow in size and computational demands, their success depends not only on algorithmic innovation but also on advances in the underlying hardware. Notably, it has been estimated that more than half of the performance gains in AI systems over the past decade can be attributed to improvements in hardware alone (Erdil & Besiroglu, 2022; Ho et al., 2024). This highlights the critical role of hardware innovation in driving and sustaining AI progress. At the same time, the growing demand for specialized and high-performance hardware places greater pressure on the efficiency of the hardware design process itself. Consequently, there is a pressing need for new methodologies that can accelerate design cycles and improve design quality and performance.

Modern hardware design is a complex, multi-stage process that begins with high-level specifications in natural language, which are manually translated into hardware description languages (HDLs) such as Verilog or VHDL, before synthesizing into circuit elements. At the heart of this process lies the register transfer level (RTL), a critical abstraction that bridges architectural intent and low-level circuit implementation, enabling designers to model intricate digital systems in a way that is both expressive and amenable to synthesis into gate-level representations. Typically, RTL design is refined iteratively, with engineers assessing the quality of the synthesized netlist using key metrics such as area and delay. However, this feedback loop is inherently slow and computationally expensive, as each iteration requires invoking a full logic synthesis toolchain. To reduce design turnaround time and improve productivity, there is an increasing demand for fast and reliable methods that can estimate design quality directly from RTL, enabling early feedback without sacrificing accuracy.

To address the aforementioned issue, prior research has explored machine learning-based approaches that represent hardware code as graphs, *e.g.*, data flow graphs (DFGs) and abstract syntax trees (ASTs), and use hand-crafted features derived from these graphs for quality estimation (Lopera et al., 2021; Sengupta et al., 2022; Fang et al., 2023). While these methods have shown encouraging results, their reliance on manually designed features significantly limits their expressiveness and ability to capture the rich structural semantics in RTL designs. As a result, these shallow representations often fail to model the complex patterns that affect downstream performance, constraining their effectiveness for accurate quality estimation. Moreover, DFGs

---

[1]Department of Computer Science and Engineering, The Chinese University of Hong Kong, Hong Kong SAR [2]Noah's Ark Lab, Huawei, Hong Kong SAR. Correspondence to: Qiang Xu <qxu@cse.cuhk.edu.hk>.

*Proceedings of the 43$^{rd}$ International Conference on Machine Learning*, Seoul, South Korea. PMLR 306, 2026. Copyright 2026 by the author(s).

provide only a partial perspective of the design by focusing solely on data dependencies while neglecting control flow, and ASTs primarily reflect syntactic hierarchy, effectively serving as an augmented form of the original code without explicitly encoding structural semantics. In contrast, control data flow graphs (CDFGs) integrate both control and data dependencies, yielding a holistic and semantically rich representation of RTL designs. This makes CDFGs particularly well suited for learning representations that are sensitive to the structural properties critical for accurate quality estimation. Recently, with the rise of large language models (LLMs) (Grattafiori et al., 2024; Hurst et al., 2024; Guo et al., 2025), some approaches have attempted to derive embeddings from RTL code using LLMs specifically trained for Verilog generation (Pei et al., 2024; Zhao et al., 2025; Liu et al., 2025a), achieving state-of-the-art performance in quality estimation tasks (Moravej et al., 2025). However, a fundamental gap exists between the objectives of code generation and quality estimation, which may limit the transferability of representations learned from generation-focused pretraining. Besides, compared to CDFG, the token-based view encodes the structural semantics of RTL designs implicitly, providing fewer cues for learning, which may hinder the final performance.

In this work, we introduce StructRTL, a novel structure-aware graph self-supervised learning framework designed to improve RTL design quality estimation. Instead of using token-based view, StructRTL operates on the CDFG view, which more explicitly captures the structural semantics of RTL designs. Besides, rather than relying on shallow, hand-crafted features, StructRTL employs two self-supervised pretraining tasks, *i.e.*, structure-aware masked node modeling and edge prediction, to learn rich structure-informed representations from CDFGs. These pretraining objectives enable the model to capture complex dependencies and design patterns, leading to significant performance improvements over existing approaches on various quality estimation tasks. Furthermore, inspired by VeriDistill (Moravej et al., 2025), we incorporate a knowledge distillation strategy that transfers low-level insights from post-mapping netlists into the CDFG-based predictor, further enhancing the model's predictive capabilities. Experimental results demonstrate that StructRTL achieves new state-of-the-art performance, highlighting the effectiveness of combining structural representation learning with cross-stage supervision. The code is open-sourced at https://github.com/cure-lab/StructRTL.

## 2. Related Work

### 2.1. RTL Design Quality Estimation

Early efforts to estimate RTL design quality have primarily relied on features extracted from graph-based representa-

tions of RTL code. For instance, Lopera et al. (2021) explore delay prediction by converting RTL designs into DFGs and extracting internal cell connectivity statistics as features to predict post-synthesis critical path delay. However, this approach has only been applied to simple combinational circuits such as adders and is trained on synthetic design variants generated by an RTL generator, limiting its practical applicability. Sengupta et al. (2022) represent RTL code as ASTs and extract hand-crafted features like total input and output bits and average wire width, which are then used to train traditional machine learning models like XGBoost (Chen & Guestrin, 2016) for quality estimation. Similarly, MasterRTL (Fang et al., 2023) converts RTL code into a bit-level design representation called the simple operator graph (SOG) and extracts features from it for the same purpose. While these approaches have shown some promise, they share two key limitations. First, their reliance on hand-crafted features limits the expressiveness of the representation and fails to capture the rich structural semantics of RTL designs. Second, their training and evaluation are conducted on small datasets, typically fewer than 100 designs in total, raising concerns about the models' generalization capabilities. Moreover, the construction of SOG requires invoking logic synthesis steps, placing it closer to the gate-level netlist stage rather than purely RTL. As such, it falls outside the scope of our work.

Recently, the emergence of LLMs (Grattafiori et al., 2024; Hurst et al., 2024; Guo et al., 2025) has sparked growing interest in fine-tuning these models for Verilog code generation (Pei et al., 2024; Zhao et al., 2025; Liu et al., 2025a;b). Following this trend, VeriDistill (Moravej et al., 2025) has proposed leveraging LLMs specifically trained for Verilog generation to extract embeddings from RTL code, achieving state-of-the-art performance on RTL quality estimation tasks using a large dataset of over 10,000 designs. However, there remains a fundamental mismatch between the objectives of code generation and quality estimation, which may limit the transferability of representations learned from generation-focused pretraining. Moreover, unlike CDFG, the token-based view encodes the structural semantics of RTL designs implicitly, offering fewer cues for learning. This lack of explicit structural information can make it more difficult for models to capture the complex patterns critical for accurate quality prediction.

### 2.2. Graph Self-Supervised Learning

Graph self-supervised learning, which learns generalizable representations from unlabeled graph data, has emerged as a popular and empirically successful learning paradigm for graph neural networks (GNNs) (Liu et al., 2022). Broadly, these methods fall into two categories: contrastive and generative approaches. Contrastive methods aim to learn representations that are invariant to different augmented views

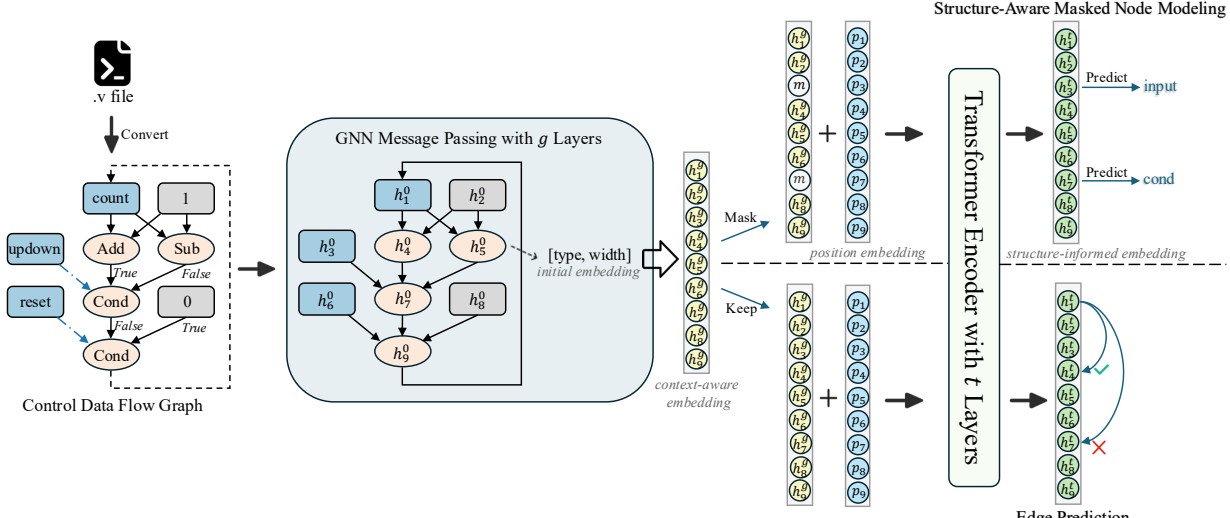

*Figure 1.* Overview of the StructRTL framework for structure-aware graph self-supervised learning. It employs two pretraining tasks: structure-aware masked node modeling, where the Transformer encoder predicts masked node types from post-GNN context-aware embeddings, and edge prediction, which recovers graph connectivity from the same embeddings without masking.

of a graph (Hassani & Khasahmadi, 2020; You et al., 2020; 2021), while generative methods create supervision signals by masking parts of the graph, such as node attributes or edges, and train the model to reconstruct the missing components (Hou et al., 2022; Li et al., 2023). In this work, we adopt generative approaches to learn structure-informed representations for RTL design quality estimation.

Inspired by the success of BERT (Devlin et al., 2019) in natural language processing (NLP), GraphMAE (Hou et al., 2022) demonstrates that masking a portion of node features and reconstructing them from their surrounding context can yield high-quality representations, achieving state-of-the-art performance on a variety of tasks. Complementarily, MaskGAE (Li et al., 2023) focuses on the structural aspect by masking a subset of edges or even paths and training the model to recover the graph connectivity, thereby learning rich topological features. These approaches compel the model to decipher the underlying relational patterns within the graph, thereby generating robust and informative embeddings. However, directly applying these techniques to CDFGs of RTL designs poses challenges. Unlike general-purpose graphs, CDFGs are tightly coupled with computational semantics, where masking nodes or edges, such as an arithmetic operator, can introduce ambiguity. For instance, if a plus operator is masked, multiple replacements (*e.g.* minus, multiply), could all appear valid, undermining the learning signal. To address this, we design two tailored pre-training objectives: structure-aware masked node modeling and edge prediction. These adaptations allow us to preserve the computational integrity of CDFGs while still leveraging the benefits of generative self-supervision. Details of these techniques are presented in the following section.

## 3. Methodology

In this section, we present the details of our proposed framework, StructRTL. It incorporates two tailored pretraining tasks, structure-aware masked node modeling and edge prediction, to learn structure-informed representations that are essential for downstream RTL quality estimation. To further improve performance, we employ a knowledge distillation strategy that transfers low-level insights from post-mapping (PM) netlists into the CDFG-based predictor, enhancing the model's ability to estimate quality metrics accurately.

### 3.1. CDFG Construction

RTL is a critical abstraction level used in digital circuit design workflow that describes the flow of data between registers and the operations performed on that data. It encompasses both combinational logic (*e.g.*, arithmetic operations and control branches) and sequential logic (*e.g.*, registers that store state across clock cycles). A common representation of RTL designs is CDFG, where nodes represent operations or storage elements, and directed edges indicate data or control dependencies between them. In sequential circuits, registers hold values between clock cycles, so their incoming edges represent inter-cycle data flow, making CDFGs cyclic graphs.

To construct a CDFG from RTL source code, we first use Yosys (Wolf et al., 2013) to compile the design into RTL intermediate language (RTLIL), a simplified form that preserves functionality while reducing the design to basic assignment and register-transfer operations. This intermediate form makes CDFG extraction more straightforward. We

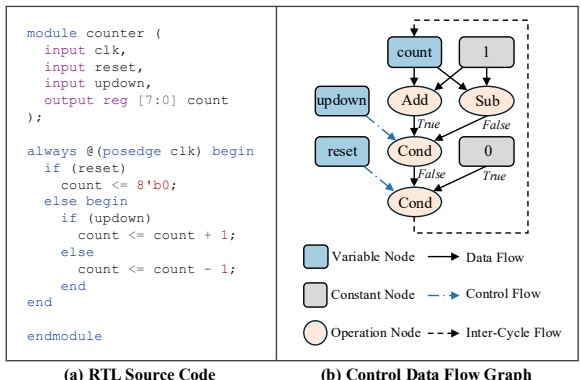

```
module counter (
  input clk,
  input reset,
  input updown,
  output reg [7:0] count
);

always @(posedge clk) begin
  if (reset)
    count <= 8'b0;
  else begin
    if (updown)
      count <= count + 1;
    else
      count <= count - 1;
    end
end

endmodule
```

**(a) RTL Source Code**      **(b) Control Data Flow Graph**

*Figure 2.* Example RTL design and corresponding CDFG.

then apply the Stagira Verilog parser (Chen et al., 2023) to generate an AST from the RTLIL. Finally, we traverse the AST to extract the CDFG. An example RTL design and its corresponding CDFG are shown in Figure 2. For more details on the CDFG, please refer to Appendix A.

### 3.2. Model Architecture

As illustrated in Figure 1, StructRTL integrates a GNN with a Transformer encoder (Vaswani et al., 2017). Given a CDFG $\mathbf{G} = (\mathbf{V}, \mathbf{E})$, where $\mathbf{V}$ denotes the set of nodes and $\mathbf{E}$ represents the edges, we first initialize the node embeddings $\{\mathbf{h}_i^0\}$ as the concatenation of the one-hot encoding of the node type and the node width:

$$\mathbf{h}_i^0 = \text{concat}\left(\text{one-hot}(\text{type}(v_i)), \text{width}(v_i)\right) \quad (1)$$

Next, message passing is performed on $\mathbf{G}$ using a graph isomorphism network (GIN) (Xu et al., 2019) to process the node embeddings and generate context-aware embeddings $\{\mathbf{h}_i^g\}$. These updated embeddings are then passed into the Transformer encoder. To preserve the structural information of the graph, we combine $\{\mathbf{h}_i^g\}$ with global positional embeddings $\{\mathbf{p}_i\}$ (Rampášek et al., 2022). This step is crucial because when the node embeddings are flattened into a sequence for the Transformer, the graph's connectivity information is lost. Without the global positional embeddings, the model would struggle to distinguish between similar nodes located in distinct regions of the graph, leading to training collapse.

To compute the global positional embeddings $\{\mathbf{p}_i\}$, we first calculate the symmetric normalized Laplacian matrix for the directed graph (Chung, 1997):

$$L = I - D_{\text{in}}^{-1/2}\left(\frac{A + A^{\text{T}}}{2}\right)D_{\text{out}}^{-1/2} \quad (2)$$

where $A$ denotes the adjacency matrix, and $D_{\text{in}}$ and $D_{\text{out}}$ are the in-degree and out-degree matrices, respectively. The

eigenvalues and corresponding eigenvectors of $L$ are then computed by solving the following equation:

$$L\mathbf{x} = \lambda\mathbf{x} \quad (3)$$

where $\{\lambda_i\}$ are the eigenvalues and $\{\mathbf{x}_i\}$ are corresponding normalized eigenvectors. To construct the global positional embeddings, we select the $k$ eigenvectors corresponding to the smallest $k$ eigenvalues, where $k$ is set to 16 in this work. If the number of nodes is smaller than 16, we pad the embeddings with $\mathbf{0}$. We then project $\{\mathbf{p}_i\}$ using a linear projection layer to ensure it has the same dimensionality as $\{\mathbf{h}_i^g\}$. Additionally, since eigenvectors $\{\mathbf{x}_i\}$ are inherently sign-insensitive (*i.e.*, both $\mathbf{x}_i$ and $-\mathbf{x}_i$ are valid eigenvectors for $\lambda_i$), we randomly flip the signs of the eigenvectors during training. This technique helps reduce the model's sensitivity to the sign of the eigenvectors and improves its generalization ability. Finally, the combined embeddings are passed into the Transformer encoder, where they are transformed into final embeddings $\{\mathbf{h}_i^t\}$ ready for downstream quality estimation via structure-aware masked node modeling and edge prediction.

### 3.3. Pretraining Tasks

In this work, we adopt two self-supervised pretraining tasks, structure-aware masked node modeling and edge prediction, to obtain structure-informed representations for downstream RTL design quality estimation. Unlike prior methods that directly masks raw node features or edges in the original graph (Hou et al., 2022; Li et al., 2023), we apply masking at the level of post-GNN context-aware embeddings. This design choice is particularly important for computational graphs like CDFGs, where each node carries strict functional semantics. Masking raw nodes or edges in such graphs can introduce ambiguity, as multiple valid replacements exist. By instead masking the post-GNN embeddings, which already encode surrounding semantics, the model can utilize rich context information from unmasked regions to reconstruct the masked content. This preserves the structural and computational integrity of the original graph, enabling more faithful reconstruction.

**Structure-Aware Masked Node Modeling.** In this task, we randomly mask 20% of the post-GNN node embeddings by replacing them with a special learnable [MASK] embedding and use the Transformer encoder to recover the masked nodes by predicting their original node types, which is formulated as a 32-class classification problem. A complete list of node types is provided in Appendix A. One key challenge in this task is the severe class imbalance. For example, operator nodes are significantly underrepresented compared to storage elements such as wires and registers. To mitigate this, we adopt two strategies. First, we apply stratified masking, which ensures that at least $m$ nodes from each class are included in the masked set during each training

---

**Algorithm 1** Stratified Masking

---

**Input:** Labels $L \in \mathbb{Z}^N$, mask ratio $r$, minimum per class $m$
**Output:** Mask $M \in \{0, 1\}^N$

1: $M \leftarrow \text{RandomBoolMask}(N, r)$ {Each entry is 1 with probability $r$}
2: **for all** class $c \in \text{Unique}(L)$ **do**
3:     $I_c \leftarrow \{i \mid L_i = c\}$
4:     $k \leftarrow \sum_{i \in I_c} M_i$ {Num. of masked nodes in class $c$}
5:     **if** $k < m$ **then**
6:         $E \leftarrow \text{RandSample}(I_c, \min(m, |I_c|))$
7:         **for all** $i \in E$ **do**
8:             $M_i \leftarrow 1$
9:         **end for**
10:    **end if**
11: **end for**
12: **return** $M$

---

**Algorithm 2** Class-Balanced Focal Loss

---

**Input:** Samples per class $S$, output logits $Z$, labels $L$
**Parameters:** $\beta = 0.9999, \gamma = 2.0$
**Output:** Loss $\mathcal{L}_{cb\_focal}$

1: **for** each class c **do**
2:     $\text{effective\_num}_c \leftarrow 1.0 - \beta^{S_c}$
3:     $w_c \leftarrow \frac{1-\beta}{\text{effective\_num}_c + 1e-8}$
4: **end for**
5: $w_c \leftarrow \frac{w_c}{\sum_c w_c} \times \text{len}(S)$
6: **for** each sample $i$ in batch **do**
7:     $ce_i \leftarrow \text{CrossEntropy}(Z_i, L_i)$
8:     $p_t \leftarrow \exp(-ce_i)$
9:     $\text{focal}_i \leftarrow (1 - p_t)^\gamma$
10:    $\text{loss}_i \leftarrow w_{c_i} \times \text{focal}_i \times ce_i$
11: **end for**
12: $\mathcal{L}_{cb\_focal} \leftarrow \frac{1}{N} \sum_{i=1}^{N} \text{loss}_i$

---

iteration. This helps the model to learn meaningful representations even for infrequent node types. The full procedure is described in Algorithm 1. Second, instead of using the classic cross entropy loss, we adopt the class-balanced focal loss (Cui et al., 2019), which adjusts the loss based on the effective number of samples for each class, and puts more focus on hard-to-classify examples. The details of the class-balanced focal loss are presented in Algorithm 2. The loss function for this task is denoted as $\mathcal{L}_{mnm} = \mathcal{L}_{cb\_focal}$.

**Edge Prediction.** When the post-GNN embeddings are flattened for input into the Transformer encoder, the graph's connectivity is lost, which can be viewed as if all edges are masked. For each training iteration, we randomly select 20% of the actual edges as positive samples and an equal number of non-existing edges as negative samples, and formulate the edge prediction task as a binary classification problem. Specifically, we concatenate the final embeddings of the source and target nodes and use a 3-layer multi-layer perceptron (MLP) to predict whether an edge exists between them. We employ cross entropy loss for this task, and denote the loss as $\mathcal{L}_{ep}$.

The total loss for pretraining is:

$$\mathcal{L}_{pre} = \alpha \mathcal{L}_{mnm} + (1 - \alpha) \mathcal{L}_{ep} \quad (4)$$

where $\alpha$ balances these two pretraining tasks. In this work, we set $\alpha = 0.5$. After pretraining, the validation accuracies for structure-aware masked node modeling and edge prediction are 82.27% and 95.77%, respectively.

### 3.4. Quality Estimation

After pretraining, StructRTL outputs structure-informed embeddings ready for RTL design quality estimation. This task is formulated as a regression problem, where node-level embeddings are first aggregated into a graph-level representation using joint mean and max pooling. The resulting graph embedding is then passed through a 3-layer MLP to predict quality metrics such as area and delay. Given that these metric values have large magnitudes and exhibit significant variance across designs, we apply a logarithm transformation to these values to make the target distribution more suitable for model learning. This transformation does not affect the practical performance of the model, as we are more concerned with the relative quality of different designs. For this task, we use the log-cosh loss (Saleh & Saleh, 2022), which is robust to outliers, and denote the loss function as $\mathcal{L}_{qe}$.

For evaluation, we use four standard regression metrics: mean absolute error (MAE), mean absolute percentage error (MAPE), coefficient of determination ($R^2$), and root relative squared error (RRSE). Given predicted values $\hat{y}_i$ and ground truth values $y_i$ for $i \in [1, N]$, these metrics are defined as:

$$\text{MAE} = \frac{1}{N} \sum_{i=1}^{N} |\hat{y}_i - y_i| \quad (5)$$

$$\text{MAPE} = \frac{1}{N} \sum_{i=1}^{N} \left| \frac{\hat{y}_i - y_i}{y_i} \right| \quad (6)$$

$$R^2 = 1 - \frac{\sum_{i=1}^{N} (\hat{y}_i - y_i)^2}{\sum_{i=1}^{N} (y_i - \bar{y})^2} \quad (7)$$

$$\text{RRSE} = \sqrt{\frac{\sum_{i=1}^{N} (\hat{y}_i - y_i)^2}{\sum_{i=1}^{N} (y_i - \bar{y})^2}} \quad (8)$$

where $\bar{y}$ denotes the mean of the ground truth values.

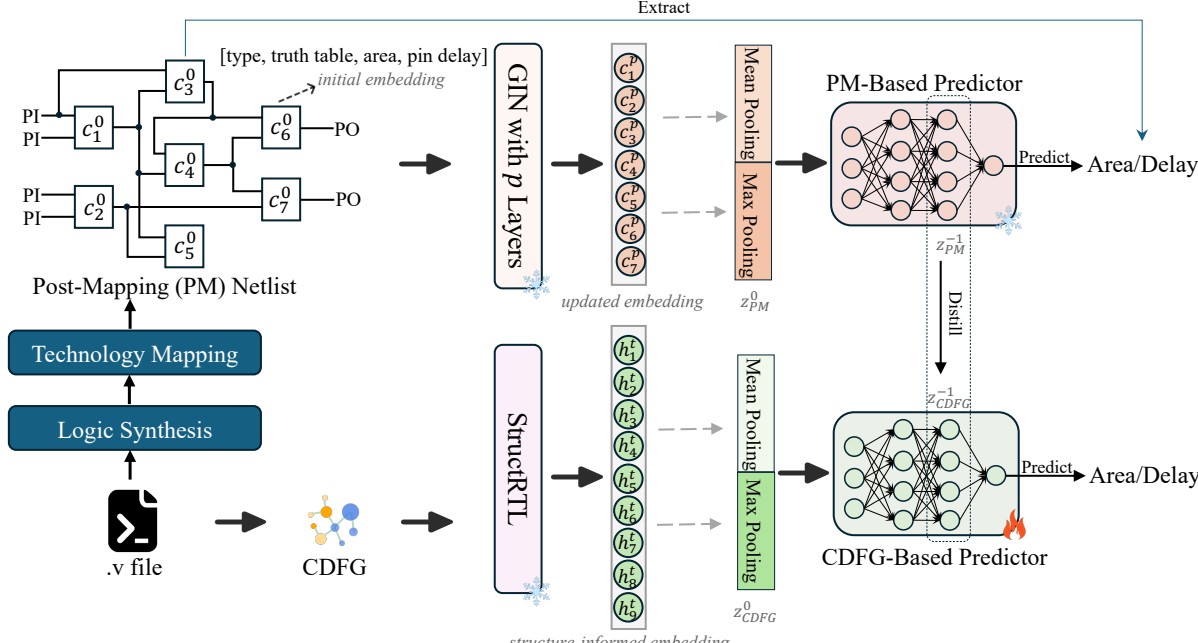

*Figure 3.* Overview of the knowledge distillation process. PM-based predictor is first trained and frozen. It then guides the learning of the CDFG-based predictor, which performs quality estimation while aligning its final-layer activations with those of the PM-based predictor.

### 3.5. Knowledge Distillation

Directly estimating quality metrics from the RTL stage can be challenging, as there remains a significant gap between RTL designs and the PM netlists from which area and delay are actually measured. To bridge this gap, we incorporate a knowledge distillation (KD) strategy that transfers low-level insights from PM netlists into the CDFG predictor. As illustrated in Figure 3, we synthesize and map RTL designs into PM netlists using Yosys (Wolf et al., 2013) and ABC (Brayton & Mishchenko, 2010) with the SkyWater 130nm technology library (Edwards, 2020). We then extract area and delay metrics from the resulting netlists. To model the PM netlist, we initialize each cell's embedding with a concatenation of its one-hot cell type encoding, logic truth table, and associated area and pin delay information. These embeddings are processed using a GIN, followed by joint mean and max pooling to obtain a graph-level representation, which is then passed to a 3-layer MLP for quality estimation. Since the area and delay values are derived directly from these netlists, the quality prediction at this stage is naturally more accurate than at the RTL level. To improve RTL-stage predictions, we perform KD by aligning the final-layer activations of the CDFG-based predictor $z_{\text{CDFG}}^{-1}$ with those of the PM-based predictor $z_{\text{PM}}^{-1}$. The KD loss is defined as:

$$\mathcal{L}_{kd} = \tau \cdot \mathcal{L}_{cos}(z_{\text{CDFG}}^{-1}, z_{\text{PM}}^{-1}) + (1 - \tau) \cdot \mathcal{L}_{mse}(z_{\text{CDFG}}^{-1}, z_{\text{PM}}^{-1}) \tag{9}$$

where $\mathcal{L}_{cos}$ is the cosine similarity loss, $\mathcal{L}_{mse}$ is the mean squared error (MSE) loss, and $\tau$ balances the contribution of the two terms. $\tau$ is set to $0.7$ in our experiments. This align-

ment encourages the CDFG-based predictor to internalize fine-grained low-level insights learned from the gate-level view, thereby improving its quality estimation performance.

During training, we first train the PM-based predictor and freeze its parameters. It is then used to guide learning of CDFG-based predictor. The total loss for training the CDFG-based predictor is:

$$\mathcal{L}_{total} = \mu \mathcal{L}_{qe} + (1 - \mu)\mathcal{L}_{kd} \tag{10}$$

where we set $\mu = 0.5$ in our experiments. The PM-based predictor is only used during training for supervision; during validation, only the CDFG-based predictor is retained.

## 4. Experiments

In this section, we conduct extensive experiments to show the effectiveness of our proposed framework, StructRTL. We begin by introducing the baselines, experimental settings, and dataset construction process. We then report comparative results on post-synthesis area and delay prediction tasks. Additionally, we perform an ablation study to demonstrate the importance of the two pretraining tasks in learning structure-informed representations. We further study StructRTL's performance under limited training data. Finally, we evaluate StructRTL on industrial designs to assess its applicability beyond open-source RTL benchmarks.

*Table 1.* Performance comparison of various methods for post-synthesis area and delay prediction without incorporating the knowledge distillation strategy. Notably, StructRTL significantly outperforms both graph-based and LLM-based baselines.

| w/o KD | Area | | | | Delay | | | |
|---|---|---|---|---|---|---|---|---|
| | MAE↓ | MAPE↓ | $R^2$↑ | RRSE↓ | MAE↓ | MAPE↓ | $R^2$↑ | RRSE↓ |
| *Graph-Based Baselines* | | | | | | | | |
| Graph-XGBoost | 0.9267 | 0.19 | 0.3987 | 0.7754 | 0.6384 | 0.12 | 0.3362 | 0.8147 |
| Graph-GNN | 0.5497 | 0.09 | 0.5857 | 0.6437 | 0.7327 | 0.13 | 0.6639 | 0.5797 |
| *LLM-Based Baselines* | | | | | | | | |
| CodeV-DS-6.7B | 0.8967 | 0.17 | 0.4862 | 0.6973 | 0.6403 | 0.12 | 0.3905 | 0.7807 |
| CodeV-CL-7B | 0.7982 | 0.15 | 0.5755 | 0.6515 | 0.5620 | 0.10 | 0.5174 | 0.6947 |
| CodeV-QW-7B | 0.7229 | 0.13 | 0.6353 | 0.6039 | 0.5340 | 0.09 | 0.5277 | 0.6872 |
| *Ours* | | | | | | | | |
| StructRTL (w/o $\mathcal{L}_{mnm}$) | 0.3900 | 0.07 | 0.7249 | 0.5245 | 0.5730 | 0.11 | 0.7473 | 0.5027 |
| StructRTL (w/o $\mathcal{L}_{ep}$) | 0.4035 | 0.07 | 0.7018 | 0.5460 | 0.5902 | 0.11 | 0.7368 | 0.5130 |
| StructRTL | 0.3649 | 0.06 | 0.7463 | 0.5037 | 0.5414 | 0.10 | 0.7630 | 0.4868 |

## 4.1. Baselines

In this work, we primarily compare StructRTL with VeriDistill (Moravej et al., 2025), a recent state-of-the-art method that leverages LLMs specifically trained for Verilog generation (Pei et al., 2024; Zhao et al., 2025; Liu et al., 2025a) to derive embeddings from RTL code. VeriDistill has shown strong performance on RTL design quality estimation tasks over a large-scale dataset of more than 10,000 designs. Following their setup, we adopt the open-source Verilog LLM CodeV (Zhao et al., 2025), which includes three variants: CodeV-DS-6.7B, CodeV-CL-7B, and CodeV-QW-7B, fine-tuned from DeepSeek-Coder (Guo et al., 2024), CodeLlama (Roziere et al., 2023), and CodeQwen (Bai et al., 2023), respectively. These models process the RTL code as raw token sequences, generate token-level embeddings, and apply mean pooling followed by a 3-layer MLP to produce final quality predictions. The LLM backbones are frozen during this process. In addition, we implement two graph-based baselines following prior work (Sengupta et al., 2022). First, we implement Graph-XGBoost, where we extract handcrafted statistical features from the CDFG, including the total bits per node type, the frequency of each node type, the average wire width, and the length of the longest combinational logic path. These features are concatenated and fed into an XGBoost (Chen & Guestrin, 2016) regressor, following the configuration described in their original work. Second, we implement Graph-GNN, which uses a GNN to directly process the CDFG without any self-supervised pretraining. The resulting node embeddings are aggregated via joint mean and max pooling and passed through a 3-layer MLP for quality estimation. This setup mirrors the architecture used in our PM-based predictor, with further details about the model presented in the following subsection.

## 4.2. Experimental Setup

This subsection details the model configurations and training hyperparameters. StructRTL uses an 8-layer GIN, followed by an 8-layer Transformer encoder, each layer containing 4 attention heads. The model is trained for 2,000 epochs with a batch size of 16, using the AdamW (Loshchilov & Hutter, 2019) optimizer with a learning rate of 2e-5 and weight decay of 1e-4. The PM-based predictor employs a 20-layer GIN with residual connections, trained for 1,000 epochs with a batch size of 256. It is optimized with the Adam (Kinga et al., 2015) optimizer at a learning rate of 1e-4 and weight decay of 1e-5. The 3-layer MLP quality estimators are trained for 600 epochs with a batch size of 256, using the same optimizer settings as the PM-based predictor. Under these configurations, all models have been trained until full convergence. The experiments are conducted on a cluster of 5 L40 GPUs, each with 46GB of memory. Further details on the model training complexity analysis can be found in Appendix B.

## 4.3. Dataset Construction

For dataset construction, we begin by collecting some existing Verilog datasets, including VeriGen (Thakur et al., 2024), RTLCoder (Liu et al., 2024), MGVerilog (Zhang et al., 2024), and DeepCircuitX (Li et al., 2025), which include designs obtained from GitHub repositories, textbook examples, and designs generated by LLMs. We filter out designs that cannot be converted into CDFGs or synthesized into PM netlists. Additionally, we use Verilator (Snyder, 2004) to simulate these designs with randomly generated input patterns, retaining only those with meaningful outputs. This ensures that the post-synthesis area and delay values

*Table 2.* Performance comparison of various methods for post-synthesis area and delay prediction with the incorporation of the knowledge distillation strategy. While knowledge distillation enhances the performance of all methods, StructRTL consistently achieves the best results, significantly outperforming the baselines, especially in delay prediction.

| w/ KD | Area | | | | Delay | | | |
|---|---|---|---|---|---|---|---|---|
| | MAE↓ | MAPE↓ | $R^2$↑ | RRSE↓ | MAE↓ | MAPE↓ | $R^2$↑ | RRSE↓ |
| *Teacher* | | | | | | | | |
| PM-based Predictor | 0.2982 | 0.05 | 0.9334 | 0.2581 | 0.1688 | 0.03 | 0.9484 | 0.2272 |
| *Graph-Based Baselines* | | | | | | | | |
| Graph-GNN | 0.4689 | 0.09 | 0.7954 | 0.4523 | 0.2926 | 0.05 | 0.8113 | 0.4344 |
| *LLM-Based Baselines* | | | | | | | | |
| CodeV-DS-6.7B | 0.4896 | 0.09 | 0.7928 | 0.4552 | 0.3787 | 0.07 | 0.7235 | 0.5258 |
| CodeV-CL-7B | 0.4192 | 0.08 | 0.8225 | 0.4213 | 0.3208 | 0.06 | 0.7696 | 0.4800 |
| CodeV-QW-7B | 0.4397 | 0.08 | 0.8174 | 0.4273 | 0.3284 | 0.06 | 0.7687 | 0.4809 |
| *Ours* | | | | | | | | |
| StructRTL (w/o $\mathcal{L}_{mnm}$) | 0.4015 | 0.07 | 0.8557 | 0.3799 | 0.2446 | 0.04 | 0.8796 | 0.3470 |
| StructRTL (w/o $\mathcal{L}_{ep}$) | 0.4071 | 0.07 | 0.8480 | 0.3899 | 0.2568 | 0.04 | 0.8654 | 0.3669 |
| StructRTL | 0.3856 | 0.07 | 0.8676 | 0.3639 | 0.2381 | 0.04 | 0.8872 | 0.3359 |

are well-defined. The resulting dataset consists of 13,200 designs, which is split into training and validation sets with a ratio of 0.8:0.2. For further details on the dataset statistics and label distribution, please refer to Appendix C.

### 4.4. Experimental Results

In this subsection, we present a comparative analysis of post-synthesis area and delay prediction between StructRTL and baseline methods. Among all evaluation metrics, we primarily focus on the $R^2$ score, as it provides insight into the relative quality of two designs, which is crucial for guiding optimization efforts. As shown in Table 1, StructRTL significantly outperforms both graph-based and LLM-based baselines without incorporating the knowledge distillation strategy, achieving a 0.1 increase in the $R^2$ score over the previous state-of-the-art methods for both area and delay prediction. Notably, Graph-XGBoost performs the worst among all methods, highlighting the limited expressiveness of hand-crafted features. Compared to Graph-GNN, which performs quality estimation in an end-to-end manner, StructRTL demonstrates superior performance, underscoring the effectiveness of structure-aware graph self-supervised learning. This approach helps the model learn structure-informed representations that are suitable and generalizable for downstream quality estimation tasks. Besides, StructRTL shows a significant advantage over LLM-based baselines, especially in the delay prediction task. This is primarily due to the fact that, unlike the token-based view, the CDFG exposes the design's structural semantics more explicitly, providing richer cues for learning complex patterns that impact the

design's quality. This is particularly important for delay prediction, which is closely tied to the design's structure. Table 2 presents the results when the knowledge distillation strategy is incorporated. As shown, knowledge distillation improves the performance of all methods, highlighting the importance of cross-stage supervision. However, StructRTL consistently achieves the best results, further validating the effectiveness of structural learning.

**Ablation Study.** We conduct an ablation study to assess the impact of the two pretraining tasks. Specifically, we remove the structure-aware masked node modeling and edge prediction, resulting in two variants: StructRTL (w/o $\mathcal{L}_{mnm}$) and StructRTL (w/o $\mathcal{L}_{ep}$), respectively. As shown in Tables 1 and 2, removing either task leads to performance degradation, demonstrating the importance of both tasks in learning the structure-aware representations necessary for accurate RTL design quality estimation. We report per-class masked node modeling results in Appendix D, provide an architectural ablation study in Appendix E, the sensitivity analysis of the edge prediction sampling ratio in Appendix F, and additional performance comparisons of StructRTL on combinational and sequential circuits in Appendix G. Experiments evaluating the generalization ability of StructRTL across functionality and design sizes are presented in Appendices H and I, respectively. The performance of LLM-based methods using CDFG representations is reported in Appendix J, whereas the performance of fine-tuning LLMs on RTL code for quality estimation is presented in Appendix K. Qualitative differences between StructRTL and LLM-based approaches are discussed in Appendix L.

## 4.5. Performance under Limited Training Data

To evaluate whether StructRTL still remains effective when only limited labeled data is available, we train StructRTL using only 5%, 10%, and 20% of the training data without the incorporation of knowledge distillation, and compare the resulting $R^2$ scores with the full-data setting and LLM-based baselines. As shown in Figure 4, StructRTL improves consistently as more labeled data becomes available. With only 20% of the training data, StructRTL already achieves competitive performance compared with full-data LLM-based baselines, reaching an $R^2$ score of 0.5603 for area prediction and 0.5973 for delay prediction. These results suggest that StructRTL can provide useful quality estimates even under a limited-data setting, thereby reducing the synthesis effort required for constructing labeled training datasets.

These findings are also relevant to the technology-node migration problem, where engineers may need to collect new labels and recalibrate quality estimation models. This requirement is a common challenge for learning-based quality estimation methods, since post-synthesis area and delay distributions can vary across technology nodes. The limited-data results above suggest that StructRTL can be adapted under such settings with a reduced amount of newly synthesized data. Moreover, since StructRTL decouples structural pretraining from quality estimation, the core structure-informed representations do not need to be learned from scratch when moving to another technology node. Developing a universal quality estimator that generalizes seamlessly across technology nodes remains an open problem.

## 4.6. Evaluation on Industrial Designs

To assess the generalization capability of StructRTL beyond open-source benchmarks, we conduct an additional evaluation on 51 industrial design pairs, totaling 102 RTL modules. Each pair consists of two functionally equivalent implementations with different post-synthesis area and delay values. These designs reflect realistic industrial implementations: each module contains over 10,000 CDFG nodes and thousands of lines of RTL code.

Since practical design space exploration often prioritizes comparing alternative implementations rather than estimating isolated quality metrics, we formulate this evaluation as a pairwise performance ranking task. Given two functionally equivalent designs, the model predicts which implementation yields superior post-synthesis quality (*i.e.*, lower area or lower delay). We use the predicted area and delay values produced by StructRTL to rank the two implementations in each pair, and evaluate the ranking accuracy against the ground-truth post-synthesis ordering. StructRTL achieves 82.35% accuracy for area ranking and 88.23% accuracy for delay ranking. These results demonstrate that the structure-aware representations learned by StructRTL remain highly

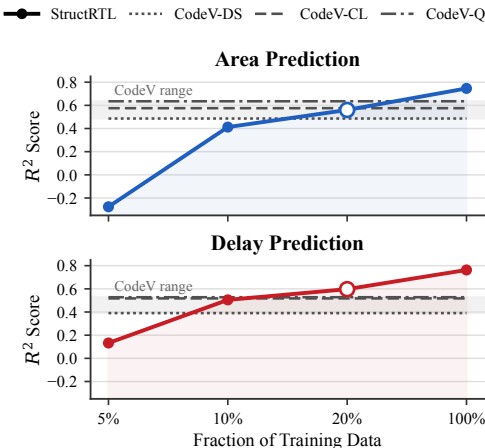

*Figure 4.* $R^2$ scores of StructRTL trained with different fractions of training data, compared with LLM-based baselines trained under the full-data setting without knowledge distillation.

effective on out-of-distribution industrial designs, highlighting its potential for practical RTL design space exploration. To further explore the scalability of StructRTL, we evaluate a register-boundary partitioning strategy on these industrial designs in Appendix M, and summarize the limitations of StructRTL in Appendix O.

**Inference Latency.** We further measure the inference latency of StructRTL on these 102 industrial modules and compare it with the corresponding synthesis time. StructRTL takes only 0.096 seconds per design on average, while synthesis takes 13.97 seconds per design on average. This $145\times$ speedup demonstrates StructRTL's ability to provide near-instantaneous quality feedback during iterative hardware design cycles. Appendix N further analyzes runtime and memory scaling with respect to graph size.

## 5. Conclusion

In this work, we introduce StructRTL, a novel structure-aware graph self-supervised learning framework to improve RTL design quality estimation. By leveraging CDFG, StructRTL incorporates two tailored pretraining tasks, structure-aware masked node modeling and edge prediction, to learn structure-informed representations for downstream quality estimation. StructRTL outperforms prior methods that either rely on shallow hand-crafted features or fail to capture the structural semantics of RTL designs. To further boost performance, we integrate a knowledge distillation strategy that transfers low-level insights from PM netlists into the CDFG-based predictor, enabling StructRTL to achieve state-of-the-art results in both area and delay prediction. Our findings highlight the effectiveness of combining structural representation learning with cross-stage supervision and open new directions for advancing RTL design quality estimation in EDA workflows.

## Acknowledgments

This work was supported in part by the Hong Kong Research Grants Council (RGC) under Grant No. 14202824, C6003-24Y, and T46-415/25-R, and in part by Huawei under Grant No. N2-2c-TH2420350.

## Impact Statement

This paper presents work whose goal is to advance the field of Machine Learning. There are many potential societal consequences of our work, none which we feel must be specifically highlighted here.

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

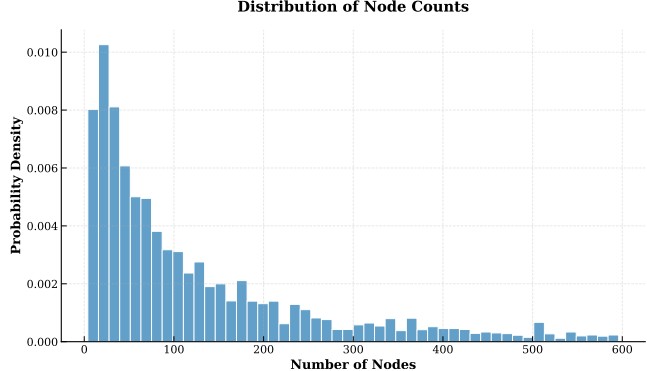

| Range | Count |
|---|---|
| $[0, 600)$ | 10645 |
| $[600, 5000)$ | 1148 |
| $[5000, 10000)$ | 542 |
| $[10000, 15000)$ | 497 |
| $[15000, 20000)$ | 368 |

*(a)* Coarse-grained distribution of node counts for all designs in the dataset.

*(b)* Fine-grained distribution of node counts for designs with fewer than 600 nodes in the CDFG.

*Figure 5.* Node count distributions in the dataset.

## A. CDFG Details

In Section 3.1, we describe the process of constructing the CDFG. In this section, we provide additional details, including a list of all node types along with their respective counts, ratios, and descriptions in our dataset, as shown in Table 12. The CDFG contains a total of 32 distinct node types, with significant variation in their frequencies. For instance, `Wire` nodes are the most common, comprising 40.05% of the dataset, while Unary Operator nodes, such as `LNot` and `Not`, make up a small portion of the dataset, with frequencies as low as 0.00% to 1.18%. For the structure-aware masked node modeling pretraining task, we formulate it as a 32-class classification problem. From Table 12, we can observe a severe class imbalance, particularly with the operation node types, which are underrepresented compared to storage elements such as `Wire` and `Reg` nodes. To mitigate this issue, we propose two strategies: stratified masking and class-balanced focal loss, which are described in the Section 3.3.

## B. Model Training Complexity Analysis

As in Section 4.2, our experiments are conducted on a cluster with five NVIDIA L40 GPUs, each equipped with 46GB of memory. Using the training setting described in Section 4.2, StructRTL can be trained to convergence on a single GPU in roughly 40 hours, with a memory footprint of about 42GB. Importantly, the batch size can be reduced to accommodate GPUs with smaller memory capacity, making the model practical for a wide range of academic and industrial users.

We also note that simpler GNN-based models, which omit the Transformer encoder and therefore have lower computational overhead, can be trained with even fewer resources. However, our experiments show that such simplified architectures experience substantial performance degradation. This highlights a fundamental trade-off: while simpler GNN-only pipelines are more lightweight, their predictive accuracy is significantly lower. StructRTL, despite its multi-stage architecture, remains a modestly sized model at only 30M parameters (114 MB in fp32), and thus is still affordable for typical research and industrial settings while delivering substantially stronger performance.

Finally, compared to LLM-based approaches, which typically require around 7B parameters, StructRTL introduces far fewer deployment barriers. Although LLM-based pipelines may appear conceptually simpler, their enormous model sizes and resource requirements make them considerably more challenging to adopt in practice than our method.

## C. Dataset Statistics

In this section, we provide additional details on the dataset statistics and label distributions. Our dataset consists of 13,200 designs, with CDFG node counts ranging from 4 to 19,169. As described in Section 4.3, the raw RTL data is sourced from four distinct Verilog datasets: VeriGen (Thakur et al., 2024), RTLCoder (Liu et al., 2024), MGVerilog (Zhang et al., 2024), and DeepCircuitX (Li et al., 2025). These datasets include designs collected from GitHub repositories, textbook examples, and designs generated by LLMs. The resulting RTL designs cover a broad spectrum of digital design functionalities that are foundational across various sectors of hardware design, from basic data processing and logical operations to complex systems

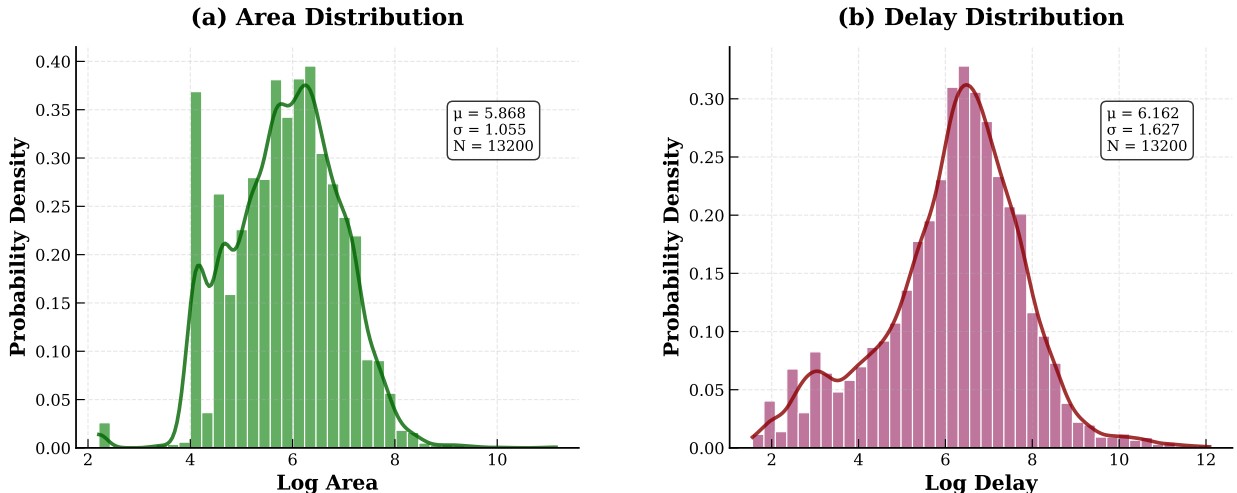

*Figure 6.* The distribution of area and delay values in the dataset, with logarithmic transformation applied to both.

involving memory management, communication interfaces, state control, multimedia applications, fault-tolerant designs, *etc.* This diversity ensures that our dataset provides a comprehensive and fair evaluation of models' quality estimation capabilities. Table 5a presents the coarse-grained distribution of node counts across the entire dataset, revealing that the majority of designs contain fewer than 600 nodes. To provide a closer look at this majority subset, Figure 5b shows a more fine-grained distribution of node counts specifically for designs with fewer than 600 nodes in the CDFG.

Figure 6 illustrates the distributions of post-synthesis area and delay values across all designs in the dataset. Given the large magnitudes and significant variance of these metrics, we apply a logarithmic transformation to normalize the target distributions and facilitate more effective model training. This transformation does not impact practical model performance, as our primary focus lies in capturing the relative quality of different designs rather than their absolute values.

## D. Per-Class Masked Node Modeling Results

As discussed in Appendix A, the CDFG node-type distribution is highly imbalanced. We report the per-class masked node modeling accuracy of StructRTL and compare it with a variant that removes the stratified masking strategy. As shown in Table 13, removing stratified masking slightly increases the overall validation accuracy from 82.27% to 82.44%. However, this improvement is mainly driven by frequent node types such as `Wire` and `Cond`, while the accuracy on rare but quality-critical operators drops substantially. For example, the accuracy on `Div`, `Mod`, and `BitNXor` decreases sharply without stratified masking. These results indicate that stratified masking prevents the pretraining task from being dominated by frequent classes and helps preserve useful representations for rare arithmetic and logic operators.

## E. Architectural Ablation Study

To isolate the contribution of each core component in StructRTL, we conduct a controlled architectural ablation study. Specifically, we compare the full StructRTL design with four variants: masking raw input features instead of post-GNN embeddings, removing positional encodings, using only the GNN without the Transformer, and using only the Transformer with positional encodings. For each setting, we report both pretraining accuracy and downstream quality estimation performance without the incorporation of knowledge distillation.

As shown in Figure 7, the full StructRTL architecture achieves the best downstream performance on both area and delay prediction. Directly masking raw input features leads to a clear performance drop because these features carry strict functional semantics, and corrupting them can introduce ambiguity that weakens the learning signal. In contrast, masking post-GNN embeddings allows the model to leverage surrounding context for reconstruction while preserving the structural and computational integrity of the original graph. Removing positional encodings causes a substantial degradation in both pretraining and downstream performance, indicating that global positional information is important once graph nodes are flattened for Transformer processing. In addition, using only the GNN or only the Transformer performs worse than the

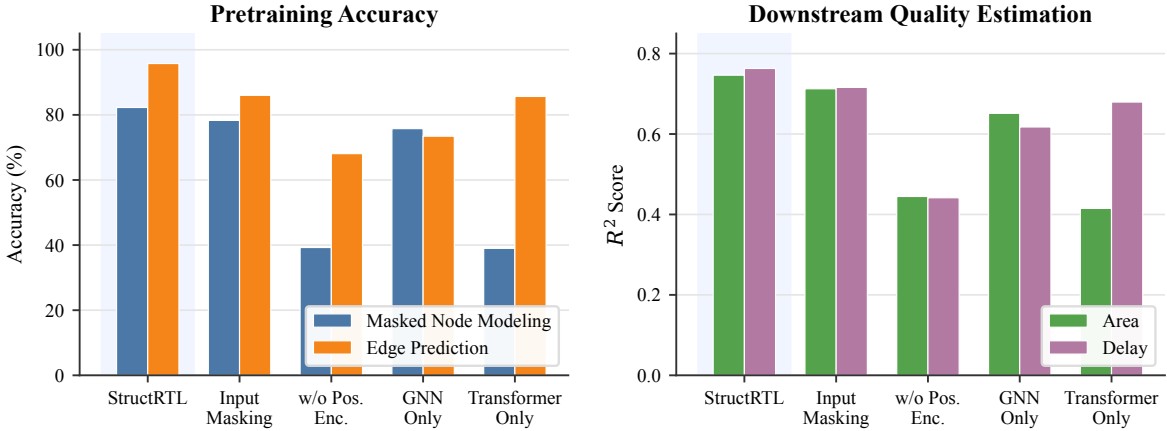

*Figure 7.* Architectural ablation of StructRTL. The full hybrid design achieves the strongest pretraining and downstream quality estimation performance, while removing positional encodings or using only one of the GNN and Transformer components leads to clear degradation.

*Table 3.* Sensitivity analysis of the edge prediction sampling ratio. We report pretraining accuracy and downstream $R^2$ scores for area and delay prediction without knowledge distillation.

| Ratio | Masked Node Modeling Acc. | Edge Pred. Acc. | Area $R^2$ | Delay $R^2$ |
|-------|---------------------------|-----------------|-----------|------------|
| 10%   | 80.91%                    | 93.82%          | 0.7448    | 0.7615     |
| 20%   | 82.27%                    | 95.77%          | 0.7463    | 0.7630     |
| 30%   | 81.17%                    | 95.89%          | 0.7459    | 0.7626     |
| 40%   | 81.08%                    | 95.98%          | 0.7455    | 0.7621     |

full hybrid model, demonstrating that local message passing and long-range dependency modeling provide complementary benefits for learning structure-informed RTL representations.

## F. Sensitivity Analysis of Edge Prediction Sampling Ratio

In the edge prediction pretraining task, we sample a subset of positive edges and the same number of negative node pairs for binary classification. To evaluate the sensitivity of StructRTL to this sampling hyperparameter, we vary the positive edge sampling ratio from 10% to 40% and report both pretraining accuracy and downstream quality estimation performance without the incorporation of knowledge distillation.

As shown in Table 3, downstream performance remains stable across different sampling ratios. The $R^2$ scores for both area and delay prediction vary only slightly, indicating that StructRTL is not highly sensitive to this hyperparameter. Among the tested settings, the 20% sampling ratio achieves the best masked node modeling accuracy and the best downstream $R^2$ scores, while also maintaining high edge prediction accuracy. Therefore, we use 20% as the default edge prediction sampling ratio in our experiments.

## G. Performance Comparison of Combinational and Sequential Circuits

In this section, we evaluate the performance difference of StructRTL on combinational and sequential circuits, providing deeper insight into its behavior. Following the original experimental setting, we partition the validation set into combinational and sequential subsets and evaluate the model on each category separately. As shown in Table 4, StructRTL generally performs better on combinational circuits than on sequential circuits across all quality estimation tasks and evaluation metrics, both with and without the incorporation of knowledge distillation. The performance gap is more pronounced for delay prediction than for area prediction. One plausible explanation is that combinational circuits typically exhibit simpler structural characteristics, as they do not involve inter-cycle data dependencies. Their CDFGs are therefore simpler and more direct, making it easier for the model to extract meaningful structural cues. In contrast, sequential circuits incorporate inter-cycle data flows and stateful elements, which complicate the underlying timing behavior and make the identification of

*Table 4.* Performance comparison of StructRTL on the combinational and sequential validation subsets, with and without the incorporation of knowledge distillation. StructRTL generally performs better on combinational circuits than on sequential circuits, with a more pronounced gap for delay prediction.

| | Area | | | | Delay | | | |
|---|---|---|---|---|---|---|---|---|
| | MAE↓ | MAPE↓ | $R^2$↑ | RRSE↓ | MAE↓ | MAPE↓ | $R^2$↑ | RRSE↓ |
| *w/o KD* | | | | | | | | |
| Combinational | 0.3579 | 0.06 | 0.7503 | 0.4997 | 0.5262 | 0.10 | 0.7923 | 0.4557 |
| Sequential | 0.3724 | 0.06 | 0.7388 | 0.5111 | 0.5562 | 0.10 | 0.7573 | 0.4926 |
| *w/ KD* | | | | | | | | |
| Combinational | 0.3753 | 0.07 | 0.8740 | 0.3550 | 0.2277 | 0.04 | 0.9002 | 0.3159 |
| Sequential | 0.3952 | 0.07 | 0.8534 | 0.3829 | 0.2478 | 0.04 | 0.8681 | 0.3632 |

*Table 5.* Comparison of the functional generalization performance of different methods for post-synthesis area and delay prediction without knowledge distillation. StructRTL consistently outperforms the LLM-based baselines.

| w/o KD (Split by Function) | Area | | | | Delay | | | |
|---|---|---|---|---|---|---|---|---|
| | MAE↓ | MAPE↓ | $R^2$↑ | RRSE↓ | MAE↓ | MAPE↓ | $R^2$↑ | RRSE↓ |
| *LLM-Based Baselines* | | | | | | | | |
| CodeV-DS-6.7B | 0.9636 | 0.20 | 0.4493 | 0.7421 | 0.6659 | 0.12 | 0.3407 | 0.8120 |
| CodeV-CL-7B | 0.8364 | 0.17 | 0.5144 | 0.6969 | 0.5854 | 0.11 | 0.4394 | 0.7487 |
| CodeV-QW-7B | 0.7840 | 0.15 | 0.5722 | 0.6541 | 0.5691 | 0.11 | 0.4505 | 0.7413 |
| *Ours* | | | | | | | | |
| StructRTL | 0.3921 | 0.07 | 0.7237 | 0.5256 | 0.5688 | 0.11 | 0.7510 | 0.4990 |

critical paths more challenging. This added complexity likely contributes to the larger performance difference observed in delay prediction, underscoring the need for more explicit modeling of temporal and stateful behaviors.

# H. Generalization Across Functionality

In this section, we evaluate the functional generalization capability of our method. Instead of using the random train/validation split adopted in the original experimental setting, we partition the dataset based on the underlying functionality of the designs. To achieve this, we first generate embeddings for each RTL design using text-embedding-3-large (Neelakantan et al., 2022) and then perform k-means clustering (Sinaga & Yang, 2020) with $k = 50$. This procedure groups functionally similar designs into the same cluster while separating those with distinct behaviors into different clusters. We then divide the dataset into training and validation sets at a ratio of 0.8:0.2 using *disjoint* clusters, thereby creating an explicit functional shift between the two sets.

As shown in Tables 5 and 6, StructRTL experiences only a slight performance drop compared with the original setting. Despite this functional shift, StructRTL maintains a strong predictive capability and continues to outperform prior state-of-the-art LLM-based baselines by a substantial margin. Furthermore, the incorporation of knowledge distillation yields additional improvements, highlighting the effectiveness of cross-stage supervision.

# I. Generalization Across Design Sizes (Scalability)

In this section, we conduct an experiment to evaluate the scalability of our method. Rather than using the random train/validation splits as in the original experimental setting, we instead partition the dataset by node count. Specifically, we use the smallest 80% of the designs as the training set, and the remaining 20% containing larger designs as the validation set. Under this configuration, the training set is composed primarily of designs with fewer than 600 nodes, whereas the validation set includes many designs with more than 10,000 nodes.

*Table 6.* Comparison of the functional generalization performance of different methods for post-synthesis area and delay prediction with knowledge distillation. StructRTL consistently outperforms the LLM-based baselines.

| w/ KD (Split by Function) | Area | | | | Delay | | | |
|---|---|---|---|---|---|---|---|---|
| | MAE↓ | MAPE↓ | $R^2$↑ | RRSE↓ | MAE↓ | MAPE↓ | $R^2$↑ | RRSE↓ |
| *Teacher* | | | | | | | | |
| PM-based Predictor | 0.3081 | 0.05 | 0.9290 | 0.2665 | 0.1928 | 0.03 | 0.9400 | 0.2450 |
| *LLM-Based Baselines* | | | | | | | | |
| CodeV-DS-6.7B | 0.5600 | 0.10 | 0.7670 | 0.4827 | 0.4067 | 0.07 | 0.6931 | 0.5540 |
| CodeV-CL-7B | 0.4374 | 0.08 | 0.7813 | 0.4677 | 0.3590 | 0.06 | 0.7200 | 0.5292 |
| CodeV-QW-7B | 0.4514 | 0.08 | 0.7769 | 0.4723 | 0.3690 | 0.06 | 0.7141 | 0.5347 |
| *Ours* | | | | | | | | |
| StructRTL | 0.3866 | 0.07 | 0.8259 | 0.4173 | 0.2768 | 0.05 | 0.8575 | 0.3775 |

*Table 7.* Comparison of the generalization capability of various methods across design sizes for post-synthesis area and delay prediction without knowledge distillation. StructRTL consistently achieves superior performance relative to LLM-based baselines.

| w/o KD (Split by Size) | Area | | | | Delay | | | |
|---|---|---|---|---|---|---|---|---|
| | MAE↓ | MAPE↓ | $R^2$↑ | RRSE↓ | MAE↓ | MAPE↓ | $R^2$↑ | RRSE↓ |
| *LLM-Based Baselines* | | | | | | | | |
| CodeV-DS-6.7B | 1.0621 | 0.22 | 0.3276 | 0.8200 | 0.6059 | 0.11 | 0.1783 | 0.9065 |
| CodeV-CL-7B | 0.9340 | 0.19 | 0.3453 | 0.8091 | 0.7133 | 0.13 | 0.1146 | 0.9410 |
| CodeV-QW-7B | 0.8724 | 0.16 | 0.4098 | 0.7682 | 0.5942 | 0.10 | 0.3267 | 0.8205 |
| *Ours* | | | | | | | | |
| StructRTL | 0.5316 | 0.09 | 0.5567 | 0.6658 | 0.5745 | 0.10 | 0.6753 | 0.5698 |

As shown in Tables 7 and 8, StructRTL exhibits a degree of performance degradation compared with the original experimental setting. Nonetheless, it still demonstrates strong generalization from small to large designs for both area and delay prediction. Its performance remains competitive and superior to prior state-of-the-art LLM-based approaches, which generally struggle to generalize beyond the scale of their training data. In addition, knowledge distillation provides further improvements, which underscores the effectiveness of cross-stage supervision.

Additionally, we observe that delay prediction generalizes better than area prediction when the model is trained on small circuits but evaluated on substantially larger ones. One plausible explanation is that area typically grows roughly proportionally with circuit size, meaning that the area values associated with very large circuits can fall far outside the distribution of values seen in smaller circuits during training, which affects generalization. In contrast, the delay of a large circuit is often dominated by the critical path within one or a few subcircuits, particularly for sequential designs. Consequently, structural patterns relevant to critical path information can be learned from small circuits and transferred effectively to larger ones. Motivated by this, we further conduct a pilot study on the register-boundary partitioning strategy in Section M, where large circuits are decomposed into subcircuits, StructRTL performs prediction on each subcircuit, and the resulting quality estimates are aggregated at the design level.

## J. LLM-Based Methods on CDFG Representation

In this section, we conduct an experiment where LLMs are used to directly process CDFG representations to further demonstrate the superiority of our method over the LLM-based ones. Importantly, the CDFG used in StructRTL is a graph-based representation that cannot be directly fed to LLMs. Therefore, for LLM-based baselines, we use the corresponding .dot-format CDFG, which provides a textual serialisation of the same underlying graph. This allows us to evaluate whether LLMs, when given the textual CDFG representation, can serve as competitive predictors of area and delay.

*Table 8.* Comparison of the generalization capability of various methods across design sizes for post-synthesis area and delay prediction with knowledge distillation. StructRTL consistently outperforms the LLM-based baselines.

| w/ KD (Split by Size) | Area | | | | Delay | | | |
|---|---|---|---|---|---|---|---|---|
| | MAE↓ | MAPE↓ | $R^2$↑ | RRSE↓ | MAE↓ | MAPE↓ | $R^2$↑ | RRSE↓ |
| ***Teacher*** | | | | | | | | |
| PM-based Predictor | 0.3628 | 0.06 | 0.8100 | 0.4359 | 0.2480 | 0.04 | 0.8966 | 0.3216 |
| ***LLM-Based Baselines*** | | | | | | | | |
| CodeV-DS-6.7B | 0.5150 | 0.09 | 0.6382 | 0.6015 | 0.3845 | 0.06 | 0.5142 | 0.6970 |
| CodeV-CL-7B | 0.5424 | 0.10 | 0.6037 | 0.6295 | 0.4260 | 0.07 | 0.4029 | 0.7727 |
| CodeV-QW-7B | 0.5024 | 0.09 | 0.6397 | 0.6002 | 0.3749 | 0.06 | 0.5444 | 0.6750 |
| ***Ours*** | | | | | | | | |
| StructRTL | 0.4441 | 0.08 | 0.6838 | 0.5623 | 0.3572 | 0.05 | 0.7443 | 0.5057 |

*Table 9.* Performance comparison of different methods for post-synthesis area and delay prediction without knowledge distillation, where LLMs directly process textualized CDFG representations. StructRTL significantly outperforms the LLM-based baselines.

| w/o KD (LLM on CDFG) | Area | | | | Delay | | | |
|---|---|---|---|---|---|---|---|---|
| | MAE↓ | MAPE↓ | $R^2$↑ | RRSE↓ | MAE↓ | MAPE↓ | $R^2$↑ | RRSE↓ |
| ***LLM-Based Baselines*** | | | | | | | | |
| CodeV-DS-6.7B | 0.9420 | 0.19 | 0.3980 | 0.7759 | 0.7260 | 0.13 | 0.3257 | 0.8212 |
| CodeV-CL-7B | 0.9039 | 0.18 | 0.4410 | 0.7477 | 0.6305 | 0.12 | 0.3671 | 0.7956 |
| CodeV-QW-7B | 0.7917 | 0.15 | 0.5639 | 0.6604 | 0.5961 | 0.11 | 0.4250 | 0.7583 |
| ***Ours*** | | | | | | | | |
| StructRTL | 0.3649 | 0.06 | 0.7463 | 0.5037 | 0.5414 | 0.10 | 0.7630 | 0.4868 |

Using these `.dot`-form CDFGs as input, we generate LLM-derived CDFG embeddings and evaluate their predictive performance under the original experimental setting. The results are summarized in Tables 9 and 10, which show that, even when replacing RTL code with CDFG inputs, StructRTL continues to outperform all LLM-based baselines by a substantial margin, both with and without knowledge distillation, on area and delay prediction.

We further observe that replacing RTL code with `.dot`-format CDFGs degrades the performance of VeriDistill. This is likely because CDFG-like representations are largely absent from the corpora used to pre-train these LLMs, leaving the models with limited prior exposure or inductive bias for such structures. As a result, their ability to extract meaningful information from textualized CDFGs is diminished, leading to weaker downstream predictive performance.

## K. Fine-Tuning LLM on RTL Code for Quality Estimation

To examine the effect of directly adapting LLM parameters for RTL quality estimation, we conduct an additional experiment using CodeV-QW-7B, the strongest LLM-based baseline in our experiments. Specifically, we fully fine-tune CodeV-QW-7B on RTL code for area and delay prediction without knowledge distillation, using the same train/validation split as the original experimental setting.

As shown in Table 11, adapting LLM parameters achieves near-perfect training performance, with $R^2$ scores of 0.9915 for area prediction and 0.9932 for delay prediction. However, its validation performance drops substantially, reaching only 0.2685 $R^2$ for area prediction and 0.0923 $R^2$ for delay prediction. This indicates severe overfitting: the 7B-parameter model can fit the training designs well, but does not generalize reliably to unseen RTL designs under the available training data. Therefore, following the setting of VeriDistill (Moravej et al., 2025), we use frozen LLMs to extract RTL representations rather than fine-tuning the full model.

*Table 10.* Performance comparison of different methods for post-synthesis area and delay prediction with knowledge distillation, where LLMs directly process textualized CDFG representations. StructRTL significantly outperforms the LLM-based baselines.

| w/ KD (LLM on CDFG) | Area | | | | Delay | | | |
|---|---|---|---|---|---|---|---|---|
| | MAE↓ | MAPE↓ | $R^2$↑ | RRSE↓ | MAE↓ | MAPE↓ | $R^2$↑ | RRSE↓ |
| ***Teacher*** | | | | | | | | |
| PM-based Predictor | 0.2982 | 0.05 | 0.9334 | 0.2581 | 0.1688 | 0.03 | 0.9484 | 0.2272 |
| ***LLM-Based Baselines*** | | | | | | | | |
| CodeV-DS-6.7B | 0.6201 | 0.11 | 0.7194 | 0.5297 | 0.4429 | 0.08 | 0.6579 | 0.5849 |
| CodeV-CL-7B | 0.5817 | 0.10 | 0.7582 | 0.4917 | 0.4140 | 0.07 | 0.7173 | 0.5317 |
| CodeV-QW-7B | 0.5826 | 0.10 | 0.7609 | 0.4890 | 0.4196 | 0.07 | 0.7084 | 0.5400 |
| ***Ours*** | | | | | | | | |
| StructRTL | 0.3856 | 0.07 | 0.8676 | 0.3639 | 0.2381 | 0.04 | 0.8872 | 0.3359 |

*Table 11.* Performance of fully fine-tuned CodeV-QW-7B on RTL quality estimation without knowledge distillation.

| Task | Set | MAE↓ | MAPE↓ | $R^2$↑ | RRSE↓ |
|---|---|---|---|---|---|
| Area | Train | 0.1104 | 0.02 | 0.9915 | 0.0921 |
| | Validation | 1.1812 | 0.21 | 0.2685 | 0.8553 |
| Delay | Train | 0.0659 | 0.01 | 0.9932 | 0.0825 |
| | Validation | 1.4844 | 0.27 | 0.0923 | 0.9527 |

## L. Qualitative Differences Between StructRTL and LLM-Based Methods

In this section, we explain why each pretraining component of StructRTL contributes to the observed performance improvements and how the resulting structural representations differ qualitatively from token-based embeddings used in LLM-based methods. StructRTL incorporates two pretraining tasks, structure-aware masked node modeling and edge prediction, to encourage the model to learn structure-informed representations from CDFGs for RTL quality estimation. Conceptually, both tasks compel the model to decipher the underlying relational patterns within the graph.

In the structure-aware masked node modeling task, we randomly mask a subset of post-GNN, context-aware node embeddings and train the Transformer to recover their original node types. Correctly predicting a masked operator node (*e.g.*, distinguishing an `Add` from a `Sub` or `Mul`) is only possible if the model has learned how that node participates in the surrounding data and control flow. This task therefore captures not only node-level semantics relevant to area (since different operator types imply differing implementation costs), but also structural characteristics relevant to delay (because operator types affect the architectural depth and critical-path behavior of the design). In essence, this task forces the model to learn both local semantics and the broader structural roles that nodes play within the CDFG.

The edge prediction task complements this by requiring the model to determine whether a connection exists between two nodes based on their learned embeddings. Recovering such connectivity forces the model to capture global structural properties (*e.g.*, reconvergence and pipeline depth) that strongly affect both area and delay. Edge prediction thus complements masked node modeling by explicitly regularizing topological understanding at the graph level.

Together, these two pretraining objectives shape the learned representations to reflect both semantic and structural characteristics of CDFGs. As shown in Tables 1 and 2, removing either pretraining task yields consistent degradation on both area and delay prediction, demonstrating that each task provides distinct yet complementary structural cues.

In contrast, token-based embeddings derived from LLMs are trained predominantly through next-token prediction on source code. This objective is highly effective for capturing syntax and high-level semantics, which accounts for LLMs' strong performance in code generation and general code understanding. However, it does not explicitly align model representations with the underlying control- or data-flow structure. Structural factors central to area and delay, such as operator connectivity, critical-path depth, or reconvergent dataflow, are only implicitly encoded in token sequences and not directly supervised during pretraining. Thus, token-based embeddings generally capture semantic content more faithfully than structural detail.

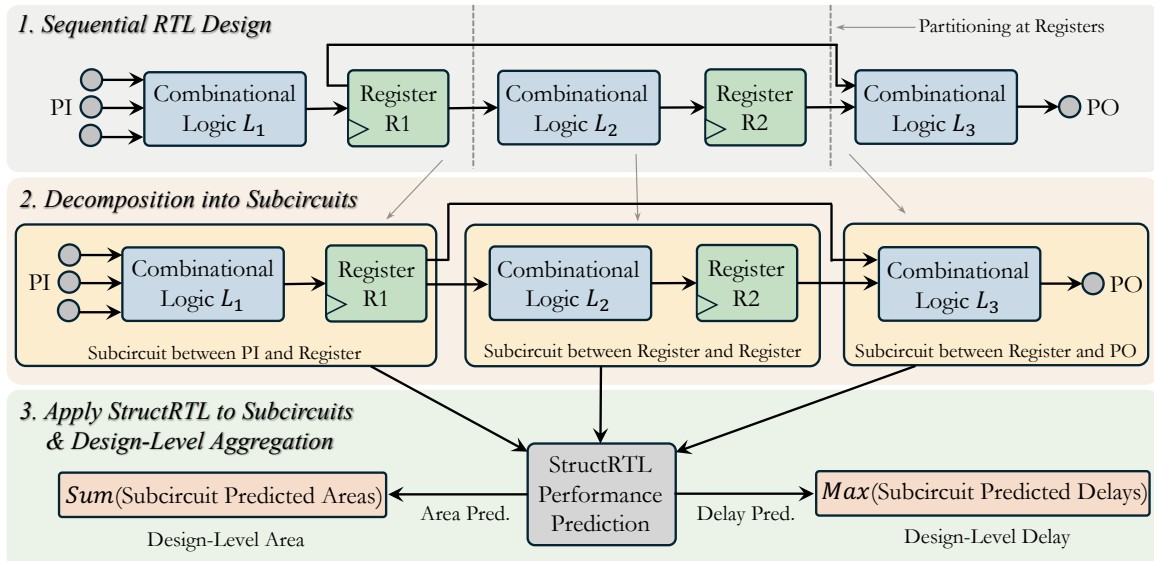

*Figure 8.* Overview of the register-boundary partitioning strategy. Sequential designs are decomposed into smaller subcircuits at register boundaries, enabling localized StructRTL inference followed by design-level aggregation.

Empirically, this distinction is clearly reflected in our results: while token-based representations somewhat narrow the gap for area prediction (which correlates more directly with node-type distributions), their performance on delay prediction remains significantly weaker. Delay is highly sensitive to structural properties that token sequences struggle to represent, whereas CDFG-based representations capture these properties explicitly. This explains why StructRTL exhibits the largest performance gains on delay, even under identical knowledge-distillation settings.

## M. Pilot Study on Register-Boundary Partitioning Strategy

To further explore the scalability of StructRTL, we conduct a pilot study using a simplified register-boundary partitioning strategy on the 102 industrial RTL modules described in Section 4.6. As illustrated in Figure 8, the key idea is to use registers as natural partition boundaries in the CDFG. This decomposes a sequential design into smaller combinational regions between primary inputs or registers and primary outputs or registers. StructRTL is then applied to the resulting subcircuits, and the subcircuit-level predictions are aggregated to obtain design-level estimates. Specifically, the overall area is estimated by summing the predicted area values of all subcircuits, while the overall delay is estimated by taking the maximum predicted delay across subcircuits. We evaluate this partitioning-based approach under the same pairwise ranking setting as in Section 4.6.

With this partitioning-based strategy, StructRTL achieves 78.43% accuracy for area ranking and 88.23% accuracy for delay ranking on the industrial design pairs, compared with 82.35% and 88.23% from the original unpartitioned setting. The delay ranking accuracy remains unchanged, while the area ranking accuracy decreases slightly. This behavior is consistent with the aggregation rule: delay depends primarily on the subcircuit containing the critical path, whereas area is obtained by summing predictions across all subcircuits and is therefore more sensitive to accumulated local errors. Although this partitioner is still preliminary and has not yet been validated on even larger designs (*e.g.*, designs with over 100,000 CDFG nodes) due to limited access to such designs, these results provide initial evidence that register-boundary partitioning is a promising direction for scaling StructRTL to larger designs.

## N. Runtime Scaling with Graph Size

To further characterize the scalability of StructRTL, we measure inference latency and peak memory usage on CDFGs with different node counts. As shown in Figure 9, inference latency increases with graph size but remains low in absolute terms. Even for designs with 20,000 CDFG nodes, StructRTL takes 201.54ms on average, providing near-instantaneous feedback compared with synthesis-based evaluation. Peak memory usage also increases with graph size, reaching 52.18GB for the

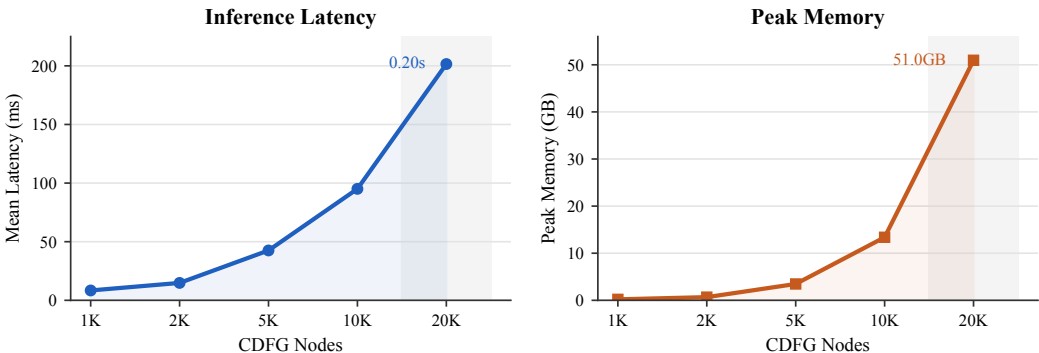

*Figure 9.* Runtime scaling of StructRTL with respect to CDFG size. We report mean inference latency and peak memory usage across designs with different node counts.

20,000-node setting. This memory growth motivates the register-boundary partitioning strategy discussed in Appendix M, which decomposes large designs into smaller subcircuits, performs localized inference on each subcircuit, and aggregates the resulting estimates at the design level. Since CDFGs explicitly expose register boundaries, such partitioning is natural in our graph-based formulation and provides a practical path toward scaling StructRTL to substantially larger designs.

## O. Limitations of StructRTL

One potential limitation of StructRTL is that model performance may degrade as circuit designs grow increasingly large and complex. Larger designs also incur higher computational cost and memory usage, which can raise the adoption barrier. However, we note that these challenges are inherent to all learning-based quality estimation methods and are not unique to our framework.

As discussed in Section M, a practical mitigation strategy is to partition large designs at register boundaries, thereby decomposing the circuit into smaller and more manageable subcircuits. Area and delay can then be estimated at the subcircuit level, with the overall area obtained by summation and the overall delay determined by taking the maximum across subcircuits. Our pilot study on industrial designs suggests that this hierarchical strategy is promising, especially for delay ranking, while the slight drop in area ranking indicates that more accurate local calibration and aggregation remain important future directions.

A notable advantage of the CDFG-based representation is that register boundaries are explicitly and naturally identifiable in the graph, making such partitioning straightforward to implement. In contrast, performing partitioning directly from Verilog source code is significantly more difficult due to syntactic and semantic complexity. From this perspective, the CDFG-based formulation provides a practical benefit over LLM-based approaches that operate solely on textual RTL representations.

Another limitation lies in the relatively constrained performance on sequential circuits, as detailed in Section G. Sequential designs introduce intricate inter-cycle data dependencies and stateful behavior, making their timing characteristics harder to infer. To address this, future work could explore sequential-specific pretraining objectives that explicitly capture temporal dependencies among registers and state transitions within the CDFG. In addition, the proposed register-boundary partitioning strategy could be extended to decompose sequential designs into localized combinational regions, enabling more accurate subcircuit-level area and delay estimation before aggregating the results.

Finally, while StructRTL can be calibrated for a target technology node using a lightweight quality estimator, developing a universal predictor that generalizes seamlessly across different technology nodes remains an open challenge. Since post-synthesis area and delay distributions can shift substantially across technologies, future work could explore technology-aware conditioning or multi-node training strategies to improve cross-node transfer.

*Table 12.* Node types in the CDFG, along with their respective counts, ratios, and descriptions.

|  | Node Type | Count | Ratio | Description |
|---|---|---|---|---|
| Unary Operator | LNot | 33521 | 1.18% | Logical NOT |
|  | Not | 24579 | 0.86% | Bitwise NOT |
|  | URxor | 119 | 0.00% | Unary reduction XOR |
|  | URand | 739 | 0.03% | Unary reduction AND |
|  | URor | 5285 | 0.19% | Unary reduction OR |
| Binary Operator | Lt | 2852 | 0.10% | Less than |
|  | Le | 2285 | 0.08% | Less than or equal |
|  | Gt | 1701 | 0.06% | Greater than |
|  | Ge | 3139 | 0.11% | Greater than or equal |
|  | Add | 30179 | 1.06% | Addition |
|  | Sub | 6177 | 0.22% | Subtraction |
|  | Mul | 3325 | 0.12% | Multiplication |
|  | Div | 198 | 0.01% | Division |
|  | Mod | 142 | 0.00% | Modulo operation |
|  | ShiftLeft | 261 | 0.01% | Logical left shift |
|  | ShiftRight | 487 | 0.02% | Logical right shift |
|  | And | 29679 | 1.04% | Logical AND |
|  | Or | 8303 | 0.29% | Logical OR |
|  | Eq | 189418 | 8.17% | Equality |
|  | Neq | 1350 | 0.05% | Inequality |
|  | BitAnd | 51656 | 1.82% | Bitwise AND |
|  | BitOr | 23330 | 0.82% | Bitwise OR |
|  | BitXor | 101147 | 3.56% | Bitwise XOR |
|  | BitNXor | 67 | 0.00% | Bitwise XNOR |
| N-ary Operator | PartSelect | 124144 | 4.36% | Bit-range extraction |
|  | Concat | 52727 | 1.85% | Bitwise concatenation |
| Cond | Cond | 571656 | 20.09% | Ternary conditional operator |
| Wire | Wire | 1139415 | 40.05% | Wire declaration or reference |
| Const | Const | 232450 | 8.17% | Constant literal value |
| I/O | Input | 73853 | 2.60% | Input port |
|  | Output | 64047 | 2.25% | Output port |
| Reg | Reg | 66680 | 2.34% | Register declaration |

*Table 13.* Per-class masked node modeling accuracy with and without stratified masking. Overall values are weighted by class frequency.

| | Node Type | Ratio | StructRTL | w/o Stratified Masking |
|---|---|---|---|---|
| | LNot | 1.18% | 68.91% | 73.04% |
| | Not | 0.86% | 64.95% | 72.37% |
| Unary Operator | URxor | 0.00% | 88.00% | 30.00% |
| | URand | 0.03% | 61.10% | 40.00% |
| | URor | 0.19% | 59.30% | 45.00% |
| | Lt | 0.10% | 62.39% | 50.00% |
| | Le | 0.08% | 59.28% | 44.00% |
| | Gt | 0.06% | 58.35% | 45.00% |
| | Ge | 0.11% | 58.20% | 45.00% |
| | Add | 1.06% | 54.43% | 54.53% |
| | Sub | 0.22% | 60.32% | 48.00% |
| | Mul | 0.12% | 63.04% | 45.45% |
| | Div | 0.01% | 57.69% | 16.67% |
| | Mod | 0.00% | 61.11% | 0.00% |
| Binary Operator | ShiftLeft | 0.01% | 56.15% | 30.00% |
| | ShiftRight | 0.02% | 58.25% | 35.00% |
| | And | 1.04% | 62.27% | 63.84% |
| | Or | 0.29% | 61.38% | 63.20% |
| | Eq | 8.17% | 84.71% | 87.77% |
| | Neq | 0.05% | 58.22% | 45.00% |
| | BitAnd | 1.82% | 62.08% | 63.20% |
| | BitOr | 0.82% | 59.94% | 52.74% |
| | BitXor | 3.56% | 90.90% | 89.81% |
| | BitNXor | 0.00% | 100.00% | 0.00% |
| N-ary Operator | PartSelect | 4.36% | 80.32% | 79.50% |
| | Concat | 1.85% | 74.35% | 67.28% |
| Cond | Cond | 20.09% | 88.91% | 88.94% |
| Wire | Wire | 40.05% | 89.67% | 90.24% |
| Const | Const | 8.17% | 67.78% | 72.04% |
| I/O | Input | 2.60% | 55.53% | 47.85% |
| | Output | 2.25% | 69.83% | 66.67% |
| Reg | Reg | 2.34% | 51.87% | 48.51% |
| | **Overall** | **100.00%** | **82.27%** | **82.44%** |

