# OpenReview forum: "Beyond Tokens: Enhancing RTL Quality Estimation via Structural Graph Learning"
_ICML.cc/2026/Conference — ICML 2026 regular_

### Official Review · Reviewer_DLKR · 2026-03-05

**Soundness:** 2
**Presentation:** 3
**Significance:** 2
**Originality:** 3
**Overall Recommendation:** 4
**Confidence:** 3

**Summary:**

The paper introduces StructRTL, a structure-aware framework for RTL quality estimation that converts designs into control/data flow graphs and models them using a GIN-Transformer hybrid with Laplacian positional encodings. The model is pretrained via masked node modeling and edge prediction, and further enhanced through knowledge distillation from post-mapping netlists to bridge the RTL-physical gap. Experiments on 13,200 real-world designs demonstrate consistent improvements over token-based, feature-based, and LLM baselines under both functional and size-based distribution shifts.

**Compliance With Llm Reviewing Policy:**

Affirmed.

**Ethical Review Concerns:**

Some of the concerns raised during the rebuttal period were not sufficiently addressed, I prefer to keep my score (3).

**Final Justification:**

The authors’ follow-up provides the missing empirical grounding for the hybrid architecture and scalability. Therefore, I am raising my overall recommendation from 3 to 4.

**Key Questions For Authors:**

Q1 - Scalability and Tractability.
How does StructRTL scale with significantly larger CDFGs in terms of nodes and edges? Is the edge prediction objective computationally tractable for industrial-scale RTL modules?

Q2 - Edge Prediction Design Choice.
Given that edges are sampled at 20% positives with equal negatives (Section 3.3), did the authors evaluate alternative sampling ratios, and how sensitive is downstream quality estimation to this hyperparameter?

Q3 - Class Imbalance and Structural Generalization.
Appendix A shows severe class imbalance among 32 node types. Could the authors provide per-class masked node accuracy, or ablations that remove stratified masking, to quantify its contribution?

The rebuttal will positively affect my evaluation if it provides additional empirical evidence (e.g., ablations or analyses) and/or a clear textual justification clarifying the contribution of the architectural components, and if it addresses the scalability and node-type imbalance concerns raised above.

**Limitations:**

Yes.

**Strengths And Weaknesses:**

Strengths
S1 - Structure-Aware Modeling.
The paper addresses a key limitation of token-level modeling by introducing graph-based structural representations tailored to RTL, which is inherently hierarchical and relational.

S2 - Pre-training Tasks.
The use of masked node modeling and edge prediction reflects a principled attempt to capture both local and relational structure in CDFGs.

S3 - Cross-Stage Knowledge Distillation.
The incorporation of knowledge distillation from post-mapping (PM) netlists (Section 3.5) is a substantial methodological contribution. By transferring gate-level insights into the CDFG-based predictor, the authors explicitly address the semantic gap between RTL and physical quality metrics.

Weaknesses
W1 - Limited Discussion of Theoretical Justification.
While the architectural components are clearly specified (Section 3.2 and Section 4.2), the paper does not provide a theoretical or empirical analysis isolating the contribution of each architectural component beyond reporting aggregate performance.

W2 - Limited Evidence of Broader Impact.
While the proposed framework is evaluated on a reasonably large RTL dataset, the experiments (Section 4) focus exclusively on the specific RTL quality estimation task. The paper does not evaluate whether the learned structural representations transfer to other EDA tasks, making it difficult to assess the approach's broader impact beyond the studied benchmark.

---

> ### Author Rebuttal · Authors · 2026-03-31
>
> ### **R1: [Architectural Component Contribution for W1]**
>
> We thank the reviewer for the thoughtful feedback. Our core architecture integrates a GNN with a Transformer encoder. This hybrid approach is necessary because computational graphs like CDFGs carry strict functional semantics. Unlike general-purpose graphs, masking raw nodes or edges in CDFGs introduces ambiguity where multiple valid replacements exist. By instead masking post-GNN embeddings, which already encode surrounding semantics, the model could utilize rich context to reconstruct content while preserving the structural and computational integrity of the original graph.
>
> This dual-module design also allows the model to capture a comprehensive local-to-global representation of the design. The GNN component serves as the local feature extractor. This is complemented by the Transformer encoder, which integrates global positional embeddings to resolve long-range dependencies across the graph. These positional embeddings are technically indispensable; without them, the Transformer fails to distinguish between similar nodes located in different regions, leading to training collapse.
>
> Furthermore, we have conducted several ablation studies throughout the paper to isolate the contributions of some architectural components:
>
> - Pretraining Tasks: Tables 1 and 2 demonstrate that removing either structure-aware masked node modeling or edge prediction leads to consistent performance degradation.
> - Cross-Stage Supervision: By comparing results with and without knowledge distillation, we isolate the significant performance boost provided by internalizing gate-level insights from post-mapping netlists.
>
> We hope these clarifications could help address the reviewer’s concerns regarding the contribution of architectural components.
>
> ### **R2: [Broader Impact for W2]**
>
> We appreciate the reviewer’s feedback regarding the broader utility of our framework. In the modern EDA workflow, verification and optimization represent the two primary bottlenecks. StructRTL specifically focuses on quality estimation, a structure-oriented task that provides the instant feedback for guiding optimization. On the other hand, verification is a function-oriented task requiring meticulous modeling of a design’s functional behavior, and this represents a separate line of research that is not the focus of our work.
>
> Furthermore, the developed framework could be transferred to other domains where graphs carry strict functional semantics, such as software program analysis and symbolic reasoning, which we believe would impact the broader machine learning community.
>
> ### **R3: [Scalability and Tractability for Q1]**
>
> We appreciate the reviewer’s questions regarding scalability. Due to character limits, we kindly refer the reviewer to our responses to Reviewer MmEh (R1), Reviewer TuFS (R2) and Reviewer ekLw (R2) for detailed discussion.
>
> ### **R4: [Edge Prediction Sampling Ratio for Q2]**
>
> We thank the reviewer for the thoughtful feedback. We conduct an ablation study across edge prediction sampling ratios of 10%, 20%, 30%, and 40% to quantify sensitivity:
>
> | **Edge Prediction Sampling Ratio** | **Masked Node Modeling Accuracy** | **Edge Prediction Accuracy** | **Area $R^2$ (w/o KD)** | **Delay** $R^2$ **(w/o KD)** |
> | --- | --- | --- | --- | --- |
> | 10% | 80.91% | 93.82% | 0.7448 | 0.7615 |
> | 20% | **82.27%** | 95.77% | **0.7463** | **0.7630** |
> | 30% | 81.17% | 95.89% | 0.7459 | 0.7626 |
> | 40% | 81.08% | **95.98%** | 0.7455 | 0.7621 |
>
> The results indicate that the downstream quality estimation performance is not highly sensitive to this hyperparameter. The 20% ratio provides the optimal trade-off, maximizing both pretraining accuracy and downstream generalization. We will incorporate these new experiments into the revised manuscript.
>
>
> ### **R5: [Class Imbalance & Stratified Masking for Q3]**
>
> We thank the reviewer for this insightful question. Removing stratified masking slightly increases overall validation accuracy (82.27% to 82.44%) because the model over-optimizes for frequent elements like Wire and Cond nodes. However, as shown below, accuracy on rare arithmetic operators, which are critical for quality estimation, collapses without this strategy:
>
> | **Node Type** | **Freq (%)** | **StructRTL** | **w/o Stratified Masking** |
> | --- | --- | --- | --- |
> | **Wire** (Common) | 39.44 | 0.8967 | **0.9024** |
> | **Cond** (Common) | 20.07 | 0.8891 | **0.8894** |
> | **Mul** (Rare) | 0.07 | **0.6304** | 0.4545 |
> | **Div** (Rare) | 0.01 | **0.5769** | 0.1667 |
> | **Mod** (Rare) | 0.01 | **0.6111** | 0.0000 |
> | **BitNXor** (Rare) | 0.00 | **1.0000** | 0.0000 |
>
> Due to character limits, we report only a representative subset of nodes here; a detailed 32-class report will be included in the revised manuscript.
>
> ---
>
> We hope our rebuttal could help address the reviewer's concerns, and we are happy to discuss further if the reviewer has any additional concerns or questions.

---

> > ### Author Rebuttal · Reviewer_DLKR · 2026-04-02
> >
> > I thank the authors for the detailed and constructive rebuttal. The additional experiments on industrial-scale designs (R3) and the new ablations on edge sampling (R4) and class imbalance (R5) are valuable and help strengthen the empirical section.
> >
> > However, a few concerns remain only partially addressed.
> >
> > **Architectural justification** The rebuttal provides useful intuition and clarifications, but it does not include controlled ablations that isolate the contributions of the core architectural components (e.g., GNN vs. Transformer vs. positional encodings). As a result, the necessity of the proposed hybrid design remains unsupported beyond qualitative reasoning.
> >
> > **Broader impact** The response remains largely speculative and does not provide empirical evidence demonstrating transferability beyond the RTL quality estimation task. This makes it difficult to assess the broader significance of the approach.
> >
> > **Scalability** While the additional industrial experiments are appreciated, the rebuttal does not provide a clear characterization of computational complexity or runtime scaling with respect to graph size, leaving open questions about tractability for significantly larger designs.
> >
> > Overall, while the rebuttal improves certain aspects of the evaluation, the remaining concerns relate to core aspects of the work (architectural justification, scalability characterization, and broader impact) and would require more substantial additions to the paper to be fully resolved.

---

> > > ### Author Response · Authors · 2026-04-08
> > >
> > > We thank the reviewer for the acknowledgement of our rebuttal and we are pleased that our responses help address several concerns. To further resolve the remaining concerns regarding architectural justification, scalability, and broader impact, we provide the following additional evidence.
> > >
> > > ### 1. Architectural Justification
> > >
> > > To isolate the contributions of our hybrid design, we conduct a new controlled ablation study. The results, summarized in the table below, demonstrate that StructRTL is a synergistic system where each component is technically indispensable for capturing the structural semantics essential for accurate RTL quality estimation.
> > >
> > > | **Setting** | **GNN** | **Transformer** | **Positional Encoding** | **Mask Raw Input** | **Masked Node Modeling Accuracy** | **Edge Prediction Accuracy** | **Area $R^2$ (w/o KD)** | **Delay** $R^2$ **(w/o KD)** |
> > > | --- | --- | --- | --- | --- | --- | --- | --- | --- |
> > > | StructRTL | Yes | Yes | Yes | No | **82.27%** | **95.77%** | **0.7463** | **0.7630** |
> > > | StructRTL (Input Masking) | Yes | Yes | Yes | Yes | 78.30% | 86.00% | 0.7125 | 0.7159 |
> > > | StructRTL (w/o Positional Encoding) | Yes | Yes | No | No | 39.26% | 68.09% | 0.4447 | 0.4415 |
> > > | GNN Only | Yes | No | No | Yes | 75.77% | 73.45% | 0.6515 | 0.6175 |
> > > | Transformer Only + Positional Encoding | No | Yes | Yes | Yes | 39.00% | 85.66% | 0.4150 | 0.6795 |
> > >
> > > **Masking Strategy:** Directly masking raw input features (Input Masking) leads to performance degradation because raw features carry strict functional semantics; masking them introduces ambiguity that undermines the learning signal. By instead masking post-GNN embeddings, the model leverages surrounding context to reconstruct content while preserving the structural and computational integrity of the original graph.
> > >
> > > **Positional Encodings:** Removing positional encodings leads to training collapse. Because graphs are flattened for the Transformer, global positional embeddings are necessary for the model to distinguish between similar nodes in different regions of the design.
> > >
> > > **Hybrid Design:** The GNN acts as a local feature extractor, while the Transformer handles long-range dependencies. As shown above, using either component in isolation fails to capture the multi-level semantics required for high-fidelity quality estimation.
> > >
> > > ### 2. Computational Complexity & Runtime Scaling
> > >
> > > We appreciate the reviewer’s insightful suggestion to characterize computational complexity and runtime scaling with respect to graph size. To quantify this, we measure the inference latency and peak memory for designs with varying node counts:
> > >
> > > | **Nodes** | **Mean Latency (ms)** | **Peak Memory (MB)** |
> > > | --- | --- | --- |
> > > | 1,000 | 8.40 | 210.9 |
> > > | 2,000 | 14.84 | 688.9 |
> > > | 5,000 | 42.53 | 3,553.4 |
> > > | 10,000 | 95.08 | 13,706.5 |
> > > | 20,000 | 201.54 | 52,184.2 |
> > >
> > > Even for designs with 20,000 nodes, StructRTL provides near-instantaneous feedback (~0.2s), offering a significant speedup over traditional synthesis toolchains. For significantly larger designs, we propose a register-boundary partitioning strategy (detailed in Appendices F and I), which decomposes complex designs into manageable subcircuits. Model inference is then performed on these subcircuits, and the final performance metrics are aggregated. As noted in our response to Reviewer TuFS (R2), this partitioning strategy is highly effective and promising for scaling to even larger graphs. Unlike raw Verilog, CDFGs naturally expose these boundaries, making partitioning straightforward and enabling localized inference that keeps memory usage within industrial GPU limits.
> > >
> > > ### 3. Broader Impact
> > >
> > > While our primary evaluation is on RTL, the core challenge of learning from semantically dense computational graphs is universal to fields like software program analysis and symbolic reasoning. Hardware CDFGs share fundamental structural properties with software data-flow graphs used in compiler optimization and code understanding. Although time constraints prevent us from conducting a full empirical evaluation on a different domain for this rebuttal—and while such analysis is outside our primary focus—we believe our methodology offers a robust, transferable paradigm for any domain involving logic-governed structures. We hope our work could serve as a foundation that the broader machine learning community can adapt to related domains.
> > >
> > > We will incorporate these comprehensive ablations and scaling analyses into the revised manuscript. We hope these could help further address the reviewer's concerns. Thank you again for your constructive guidance throughout the review process.

---

### Official Review · Reviewer_ekLw · 2026-03-11

**Soundness:** 2
**Presentation:** 3
**Significance:** 2
**Originality:** 2
**Overall Recommendation:** 3
**Confidence:** 4

**Summary:**

The paper proposes StructRTL, a structure-aware graph self-supervised learning framework for estimating RTL design quality. Unlike recent LLM-based approaches that treat code as token sequences, this method converts RTL into Control Data Flow Graphs to explicitly capture structural features. The framework utilizes a GNN combined with a Transformer encoder and introduces two pretraining tasks: structure-aware masked node modeling and edge prediction. Furthermore, it employs a knowledge distillation strategy.

**Compliance With Llm Reviewing Policy:**

Affirmed.

**Final Justification:**

The rebuttal has not modified my fundamental view of the study’s quality. I am thus maintaining the original score.

**Key Questions For Authors:**

1. The Laplacian eigenvector computation is expensive for large graphs. How does your method handle the computational overhead and memory consumption when scaling to industrial-level designs? Have you evaluated the inference latency on such large graphs compared to fast synthesis modes in commercial tools?
2. The majority of the validation set consists of small designs (<600 nodes). Large circuits often exhibit different structural properties compared to small ones. How can you ensure the model generalizes to large-scale designs when the training distribution is so heavily skewed towards small circuits?
3. The knowledge distillation strategy relies on a teacher model trained on post-mapping netlists. Since synthesis is slow, does the cost of constructing the dataset and training the teacher model outweigh the benefits of the student model's inference speed, especially considering that the model might need retraining for different technology nodes (e.g., moving from 130nm to 7nm)?

**Limitations:**

yes.

**Strengths And Weaknesses:**

Strengths
1. Presentation: The paper is generally well-written, well-organized, and easy to follow. The motivation for using structural graphs over token-based views is clearly articulated.
2. Performance: The method achieves SOTA results on the constructed dataset, showing significant improvements over LLM-based baselines.
3. Comprehensive Evaluation: The experimental section is thorough, including detailed metrics for Area/Delay prediction, ablation studies verifying the contribution of pretraining tasks, and analysis of generalization across different design functionalities.

Weaknesses
1. Limited Technical Novelty: The proposed framework appears to be a combination of existing techniques (Graph Masked Modeling and Knowledge Distillation) applied to the RTL context. While the application is relevant, the work does not introduce fundamentally new learning principles or graph theory innovations.
2. Scalability Concerns: The reliance on Laplacian positional encodings involves eigendecomposition. This makes the approach computationally prohibitive for large-scale industrial graphs. Moreover, The knowledge distillation process requires a "teacher" model trained on synthesized netlists. Generating ground truth for large-scale datasets via synthesis is extremely time-consuming, potentially negating the "fast estimation" benefit during the model development phase.
3. Although the dataset contains a large number of designs (~13k), the complexity of these designs is low. As shown in Appendix C, over 80% of the designs have fewer than 600 nodes in their CDFG. This "toy-level" scale casts doubt on the model's ability to generalize to complex, real-world industrial circuits.

---

> ### Author Rebuttal · Authors · 2026-03-30
>
> ### **R1: [Technical Novelty for W1]**
>
> We thank the reviewer for the thoughtful feedback. Due to character constraints for the rebuttal, we kindly refer the reviewer to our detailed responses to Reviewer MmEh (R3) and Reviewer TuFS (R1), which address concerns regarding technical novelty and transferable ML insights.
>
> ### **R2: [Scalability & Industrial Generalization & Inference Latency for W2, W3, Q1 & Q2]**
>
> We thank the reviewer for the insightful feedback. In Appendix F, we conduct an experiment to evaluate the scalability of our method by training StructRTL on the smallest 80% of designs (primarily <600 nodes) and validating it on the largest 20%, which includes complex circuits with over 10,000 nodes. The results demonstrate that StructRTL maintains strong predictive performance and consistently outperforms LLM-based baselines, which often struggle to generalize beyond their training scale.
>
> The generalization capability is rooted in the hierarchical nature of hardware design: large circuits are typically composed of reusable functional components with similar structures. By internalizing these fundamental primitives from smaller designs, StructRTL could identify the structural patterns and critical paths that dominate the quality metrics of significantly larger systems.
>
> To further address concerns regarding industrial applicability and scalability, we kindly refer the reviewer to our responses to Reviewer MmEh (R1) and Reviewer TuFS (R2) for detailed evidence:
>
> - **Industrial Evaluation (MmEh R1):** We demonstrate effective generalization on 102 industrial modules with high pairwise ranking accuracy. On these modules, StructRTL achieves an average inference time of 0.096s per design, whereas fast synthesis takes 13.97s. This 145x speedup underscores our model’s superiority in providing near-instantaneous feedback essential for accelerating iterative hardware design cycles.
> - **Partitioning Strategy (TuFS R2):** We validate the register-boundary partitioning approach (proposed in Appendices F and I) that successfully decomposes large designs into manageable subcircuits, showing high promise for scaling to even larger designs. This strategy could be directly utilized to mitigate the computational overhead of Laplacian eigenvector computation on large graphs by ensuring eigendecomposition is performed only on smaller, localized sub-graphs.
>
> We hope this could help address the reviewer’s concerns, and we will incorporate these new findings into the revised manuscript.
>
> ### **R3: [Dataset & Training Overhead for W2 & Q3]**
>
> We understand the reviewer for these practical concerns regarding the cost of dataset construction and model training. We would like to clarify that logic synthesis and teacher model training are one-time efforts during the development phase. As noted in our response to Reviewer MmEh (R5), at inference time, StructRTL operates purely on the RTL-derived CDFG, ensuring fast estimation. Furthermore, generating ground-truth labels for any supervised quality estimation model always requires synthesis; once these labels exist, training our teacher (GIN+MLP) and student (MLP) models is computationally inexpensive compared to the synthesis itself.
>
> To address the "time-consuming" nature of dataset construction, we conduct an experiment (without KD) to evaluate performance under limited training data:
>
> **Table 1: Area Prediction Performance**
>
> | **Training Data** | **MAE$\downarrow$** | **MAPE$\downarrow$** | **R2$\uparrow$** | **RRSE$\downarrow$** |
> | --- | --- | --- | --- | --- |
> | 5% | 0.9788 | 0.16 | -0.2757 | 1.1295 |
> | 10% | 0.5911 | 0.10 | 0.4124 | 0.7666 |
> | 20% | 0.4983 | 0.09 | 0.5603 | 0.6631 |
> | 100% | 0.3649 | 0.06 | 0.7463 | 0.5037 |
>
> **Table 2: Delay Prediction Performance**
>
> | **Training Data** | **MAE$\downarrow$** | **MAPE$\downarrow$** | **R2$\uparrow$** | **RRSE$\downarrow$** |
> | --- | --- | --- | --- | --- |
> | 5% | 1.2057 | 0.20 | 0.1324 | 0.9315 |
> | 10% | 0.8360 | 0.15 | 0.5048 | 0.7037 |
> | 20% | 0.7376 | 0.14 | 0.5973 | 0.6346 |
> | 100% | 0.5414 | 0.10 | 0.7630 | 0.4868 |
>
> Even with only 20% of the training data, StructRTL achieves $R^2$ scores comparable to full-scale LLM-based baselines like CodeV-QW-7B. This demonstrates that our model can become effective with significantly reduced synthesis overhead.
>
> In industrial workflows, design teams typically target a specific technology node for the duration of a project's development cycle, making node-specific training a manageable one-time setup cost. Crucially, because StructRTL decouples structural pretraining from quality estimation, the core structure-informed embeddings do not need to be retrained from scratch when moving from 130nm to 7nm; only the lightweight MLP-based predictor requires calibration.
>
> We agree that developing a universal model across different technology nodes remains an open challenge. We will highlight this in our revised Limitations section and incorporate these findings into the manuscript.

---

> > ### Author Rebuttal · Reviewer_ekLw · 2026-04-03
> >
> > While the authors’ response has clarified several points and responded to some of my concerns, the empirical completeness of the work remains insufficiently strengthened. I select (a) with no additional questions and keep my original overall assessment.

---

> > > ### Author Response · Authors · 2026-04-08
> > >
> > > We thank the reviewer for the acknowledgement of our rebuttal and we are pleased that our rebuttal is helpful in addressing the reviewer’s concerns. To further strengthen the empirical completeness of our work, we will incorporate the new industrial evaluation, inference latency analysis, and limited-training-data study into the revised manuscript. We will also integrate the register-boundary partition pilot study and update the Limitations section to explicitly reflect current evaluation boundaries and open challenges. Thank you again for your constructive guidance throughout the review process.

---

### Official Review · Reviewer_TuFS · 2026-03-12

**Soundness:** 3
**Presentation:** 3
**Significance:** 3
**Originality:** 2
**Overall Recommendation:** 4
**Confidence:** 4

**Summary:**

StructRTL learns structure-informed representations from control data flow graphs (CDFGs) for RTL design quality estimation. It combines a GIN encoder with a Transformer, pretrained via structure-aware masked node modeling and edge prediction, followed by fine-tuning for area/delay prediction. A knowledge distillation strategy transfers insights from post-mapping netlists. On a 13,200-design dataset, StructRTL achieves R² = 0.87 (area) and 0.89 (delay) with KD, outperforming LLM-based baselines.

**Compliance With Llm Reviewing Policy:**

Affirmed.

**Final Justification:**

The rebuttal solved all my main concerns, so I will change my score from 3 to 4.

**Key Questions For Authors:**

1) What fundamental ML contribution does this work make beyond applying existing graph SSL techniques (GraphMAE, MaskGAE) to a new domain? Can you articulate a transferable insight?

2) Can you demonstrate performance on industrial-scale designs (>100k nodes) to address the scalability and representativeness concern?

3) The register-boundary partitioning idea (mentioned in Appendix I) could address scalability—has any progress been made on this direction?

4) How does StructRTL compare to fine-tuning LLMs (not just using frozen embeddings) on RTL code for quality estimation?

5) Given that KD requires running synthesis, what is the practical advantage over simply using the post-mapping predictor directly?

**Limitations:**

yes

**Strengths And Weaknesses:**

Strengths

1) The experiments are thorough: main results with/without KD, ablation of each pretraining task, combinational vs. sequential analysis (Appendix D), cross-functionality generalization (Appendix E), cross-scale generalization (Appendix F), and LLM-on-CDFG comparison (Appendix G). This is a commendably comprehensive experimental suite.

2) The argument that CDFG captures structural semantics more explicitly than token-based LLM representations is well-supported, particularly by the delay prediction results where the structural advantage is most pronounced (R² improvement from 0.53 to 0.76 w/o KD).

Weaknesses

1) The core methodology largely builds on existing representation learning techniques with limited fundamental novelty. Masked node modeling derives from GraphMAE, edge prediction from MaskGAE, and knowledge distillation from VeriDistill. The domain-specific adaptations (stratified masking, post-GNN masking, class-balanced focal loss) are reasonable engineering refinements but do not constitute methodological innovations.

2) The dataset (13,200 designs from public sources) lacks industrial representativeness. 80% of designs have fewer than 600 nodes. No industrial-scale designs are included. It remains unclear whether the method generalizes to real-world complexity.

3) CDFG construction requires Yosys compilation, and KD requires full logic synthesis + technology mapping. This contradicts the "directly from RTL" positioning. The pipeline complexity (CDFG extraction → GNN → Transformer → KD from synthesized netlists) creates adoption barriers relative to simpler approaches.

4) The contributions are primarily application-driven within EDA, offering limited theoretical or methodological advances that would engage the broader ML research community. It is unclear what fundamental ML insight StructRTL provides beyond demonstrating that graph SSL works on CDFGs.

---

> ### Author Rebuttal · Authors · 2026-03-30
>
> ### **R1: [Novelty & ML Contribution for W1, W4 & Q1]**
> As in our response to Reviewer MmEh (R3), the novelty of StructRTL lies not in its individual components, but in the synergistic system tailored to capture the structural semantics essential for RTL quality estimation.
>
> Our work offers two primary transferable insights. First, for logic-governed computational graphs where nodes carry strict functional semantics, masking raw features introduces ambiguity because multiple valid replacements often exist. Instead, masking post-GNN context-aware embeddings allows the model to leverage surrounding semantics for reconstruction while preserving the structural and computational integrity of the original graph. Second, we provide a robust methodology for handling the extreme class imbalance inherent in program-derived graphs through the integration of stratified masking and class-balanced focal loss.
>
> We believe these paradigms for learning representations from semantically dense graphs could impact the broader ML community in areas such as software program analysis and symbolic reasoning. We will update the manuscript to highlight these contributions and their potential for cross-disciplinary impact.
>
> ### **R2: [Industrial Applicability & Partitioning Strategy for W2, Q2 & Q3]**
>
> Regarding industrial applicability, we kindly refer the reviewer to our response to Reviewer MmEh (R1), where we evaluate StructRTL on 51 industrial design pairs provided by an industrial collaborator. While these industrial designs are significantly larger than the majority of our current dataset, we acknowledge that they do not exceed 100,000 nodes due to limited access to such designs; we will explicitly highlight this boundary in the revised Limitations section.
>
> Furthermore, we have conducted a pilot study to validate the register-boundary partitioning approach. Using a simplified version of this partitioner on the 102 industrial modules, we achieve 78.43% accuracy for area and 88.23% accuracy for delay (compared to 82.35% and 88.23% originally). These results demonstrate that the partitioning strategy is highly promising for handling larger designs. The performance remains robust for delay because it is determined by the maximum path; as long as the subcircuit containing the critical path is correctly identified, the global estimate is preserved. The slight drop in area accuracy is expected because area is additive, where small localized estimation errors in each subcircuit compound during the final summation.
>
> Combined with our scalability generalization experiments in Appendix F, we hope this helps mitigate the reviewer’s concern regarding the scalability and industrial applicability of StructRTL. We will incorporate these findings into the revised manuscript.
>
> ### **R3: [Practical Advantage for W3 & Q5]**
>
> Regarding the practical advantage of StructRTL, we emphasize that the teacher model and logic synthesis are strictly confined to the training phase. As noted in our response to Reviewer MmEh (R5), the PM-based predictor is discarded after training; at inference time, StructRTL operates purely on the RTL-derived CDFG.
>
> Furthermore, we would like to clarify that the use of Yosys for CDFG construction does not contradict our “directly from RTL” positioning. It is utilized solely for a near-instantaneous conversion into the RTLIL intermediate representation, not for heavy logic optimization. To address concerns regarding pipeline complexity, we provide an automated end-to-end script in the supplementary materials for the reviewer’s reference. Finally, with only 30M parameters, StructRTL is significantly more resource-efficient and easier to deploy than the 7B-parameter LLM baselines.
>
> ### **R4: [Comparison with LLM Fine-Tuning for Q4]**
>
> We thank the reviewer for this insightful question regarding the impact of LLM fine-tuning. To address this, we conduct an experiment using CodeV-QW-7B, our strongest LLM baseline, and perform full fine-tuning on RTL code for quality estimation without knowledge distillation.
>
> **Table 1: Area Prediction Performance**
>
> | **Method** | **Set** | **MAE$\downarrow$** | **MAPE$\downarrow$** | **R2$\uparrow$** | **RRSE$\downarrow$** |
> | --- | --- | --- | --- | --- | --- |
> | CodeV-QW-7B (Full FT) | Train | 0.1104 | 0.02 | 0.9915 | 0.0921 |
> | CodeV-QW-7B (Full FT) | Validation | 1.1812 | 0.21 | 0.2685 | 0.8553 |
>
> **Table 2: Delay Prediction Performance**
>
> | **Method** | **Set** | **MAE$\downarrow$** | **MAPE$\downarrow$** | **R2$\uparrow$** | **RRSE$\downarrow$** |
> | --- | --- | --- | --- | --- | --- |
> | CodeV-QW-7B (Full FT) | Train | 0.0659 | 0.01 | 0.9932 | 0.0825 |
> | CodeV-QW-7B (Full FT) | Validation | 1.4844 | 0.27 | 0.0923 | 0.9527 |
>
> The results reveal catastrophic overfitting: while the 7B-parameter model achieves near-perfect training accuracy, its validation performance drops significantly compared to the frozen baseline. We will incorporate these findings into the revised manuscript.

---

> > ### Author Rebuttal · Reviewer_TuFS · 2026-04-04
> >
> > Thanks for your great answers. I will consider raising my score from 3 to 4 in the final decision.

---

> > > ### Author Response · Authors · 2026-04-08
> > >
> > > We thank the reviewer for the acknowledgement of our rebuttal and for the positive reconsideration of our work. We are pleased that our responses help address the reviewer’s concerns. In the revised manuscript, we will incorporate the industrial design evaluation, the register-boundary partitioning pilot study, and the LLM fine-tuning analysis. We will also update the Limitations section to explicitly highlight the current scalability boundaries. Thank you again for your constructive guidance throughout the review process.

---

### Official Review · Reviewer_MmEh · 2026-03-13

**Soundness:** 3
**Presentation:** 3
**Significance:** 3
**Originality:** 3
**Overall Recommendation:** 5
**Confidence:** 3

**Summary:**

The paper studies fast prediction of hardware design quality metrics (area and delay) directly from RTL code, aiming to reduce the need for expensive synthesis iterations in electronic design automation workflows. It proposes StructRTL, a structure-aware graph self-supervised framework that operates on control-data flow graphs (CDFGs) extracted from RTL designs.

**Compliance With Llm Reviewing Policy:**

Affirmed.

**Key Questions For Authors:**

NA

**Limitations:**

Yes

**Strengths And Weaknesses:**

Strenght:
Well-motivated problem with clear practical importance in hardware design acceleration.
Strong intuition: structural graph representations align better with circuit timing and resource behavior than token embeddings.
Methodologically solid combination of graph SSL + transformer modeling + distillation.
Empirical evaluation includes ablations showing both pretraining tasks contribute to performance gains.
Model is relatively compact (~30M parameters) compared to LLM baselines.

Weaknesses:
Evaluation remains limited to academic-scale RTL datasets; real industrial designs may differ substantially in size and complexity.
Graph extraction relies on EDA tooling assumptions, which could introduce pipeline fragility.
Gains over strong LLM baselines are convincing but architectural novelty is incremental relative to existing graph SSL and graph-transformer literature.
Sequential circuit performance remains weaker, suggesting limited modeling of temporal/stateful behavior.
Knowledge distillation depends on availability of synthesized netlists during training, partially reducing the “early-prediction” advantage.

---

> ### Author Rebuttal · Authors · 2026-03-30
>
> ### **R1: [Evaluation on Industrial Designs]**
>
> We thank the reviewer for the thoughtful feedback. To address this concern, we conduct additional experiments on 51 industrial design pairs (totalling 102 modules) provided by an industrial collaborator. These pairs consist of functionally equivalent designs with different area and delay values and each module contains over 10,000 CDFG nodes and thousands of lines of RTL code. For this evaluation, we utilize a pairwise performance ranking task, which is a critical real-world use case in design space exploration where the model must predict which of two equivalent designs achieves superior performance. StructRTL seamlessly adapts to this ranking task, achieving 82.35% accuracy for area and 88.23% accuracy for delay prediction. These results demonstrate that StructRTL generalizes effectively to “out-of-distribution” industrial designs. We hope these findings could help mitigate the reviewer’s concern regarding industrial applicability and will incorporate this new evaluation into the revised manuscript.
>
> ### **R2: [Graph Extraction & Pipeline Robustness]**
>
> We thank the reviewer for this practical concern regarding pipeline robustness. StructRTL utilizes Yosys and Stagira Verilog parser to construct the CDFG. This dependency serves as an inherent preliminary test for synthesizability: designs that cannot be successfully converted into a CDFG typically fail to meet the syntactic requirements for logic synthesis. Since the task of quality estimation is only valid for synthesizable designs, this graph extraction process acts as a necessary filter to ensure the model focuses on viable hardware implementations.
>
> ### **R3: [Architectural Novelty]**
>
> We thank the reviewer for recognizing that our gains over strong LLM baselines are convincing. We would like to clarify that the novelty of StructRTL lies not in its components, but in the synergistic system tailored to capture the structural semantics essential for RTL quality estimation. Unlike general-purpose graphs, computational graphs like CDFGs carry strict functional semantics where masking raw nodes or edges introduces ambiguity, as multiple valid replacements exist. By instead masking post-GNN embeddings, the model could utilize rich context from unmasked regions to reconstruct content while preserving the structural and computational integrity of the original graph.
>
> Additionally, we address the severe class imbalance between rare operator nodes and common storage elements through stratified masking and class-balanced focal loss. We believe these paradigms for learning representations from semantically dense computational graphs could impact broader machine learning communities, including software program analysis and symbolic reasoning. We will update the manuscript to highlight these contributions and their potential for cross-disciplinary impact.
>
> ### **R4: [Sequential Performance]**
>
> We thank the reviewer for this insightful observation and acknowledge that sequential circuit performance is currently a limitation of our framework.
>
> As discussed in Appendix D, sequential circuits incorporate inter-cycle data dependencies that significantly complicate timing behavior and make the identification of critical paths more challenging compared to combinational designs. To address this, we propose two potential directions for improvement:
>
> - Hierarchical Partitioning: A partitioning strategy at register boundaries could be implemented to decompose complex sequential designs into manageable combinational subcircuits, enabling localized and more accurate area and delay estimation.
> - Sequential Pretraining: New pretraining objectives could be defined to specifically capture temporal and stateful behaviors within the CDFG, helping the model more explicitly internalize the complex dependencies inherent in sequential logic.
>
> We appreciate this valuable feedback and will update the “Limitations and Future Work” section of the manuscript to incorporate these points.
>
> ### **R5: [Early-Prediction Advantage]**
>
> We thank the reviewer for this insightful observation. We would like to clarify that the “early-prediction” advantage of StructRTL is fully preserved, as synthesized netlists and the teacher model are only utilized during the training phase. During this stage, we synthesize RTL designs into post-mapping (PM) netlists once to provide high-fidelity supervision signals. This allows the CDFG-based predictor to internalize fine-grained, gate-level insights through the knowledge distillation strategy.
>
> Crucially, the PM-based predictor is used only for supervision during training; during validation and actual use, it is discarded. At inference time, StructRTL operates purely on the RTL-derived CDFG without requiring time-consuming logic synthesis or technology mapping.
>
> ---
>
> We are happy to discuss further if the reviewer has any additional concerns or questions.

---

> > ### Author Rebuttal · Reviewer_MmEh · 2026-04-02
> >
> > I think the authors for their rebuttal.
> > I chose (a) because I do not have further questions, yet most the weaknesses of this works remains as is, as the authors also mentioned. Nevertheless, I am convinced of the potential contribution of the proposed approach to the community despite its limited current scope, therefore I maintain my positive score which also reflects the weaknesses.

---

> > > ### Author Response · Authors · 2026-04-08
> > >
> > > We thank the reviewer for the acknowledgement of our rebuttal and for the positive feedback regarding the potential contribution of our work to the community. We acknowledge the current limitations of our framework (e.g., the performance gap observed in sequential circuits), and will update the Limitations section to explicitly highlight these challenges alongside potential solutions, including hierarchical subcircuit partitioning at register boundaries and the development of temporal-aware pretraining objectives. Furthermore, we will integrate the new industrial design experiments into the revised manuscript to provide a more comprehensive view of the model’s scalability and real-world applicability. Finally, we hope that our method could serve the broader machine learning community and be transferred to other domains like software analysis and symbolic reasoning for analyzing semantically dense computational graphs. Thank you again for your constructive review and for maintaining your positive score.

---

### Decision · Program_Chairs · 2026-04-30

**Decision:**

Accept (regular)

**Comment:**

Reviews thought that this was an exciting problem area and were generally positive on the paper.  However several would have appreciated experiments on larger or more realistic designs.

Strengths:
- important problem, and sound solution design
- model is more efficient than LLM solutions

Weaknesses:
-  experimentation on larger, more complicated designs (which might have different structural characteristics) is lacking
-  architecture novelty is limited compared to existing work
-  scalability concerns (which could be addressed by experiments on larger designs)